# Deep-sequence phylogenetics to quantify patterns of HIV transmission in the context of a universal testing and treatment trial – BCPP/Ya Tsie trial

Lerato E Magosi[1]*, Yinfeng Zhang[2], Tanya Golubchik[3], Victor DeGruttola[4], Eric Tchetgen Tchetgen[5], Vladimir Novitsky[6,7], Janet Moore[8], Pam Bachanas[8], Tebogo Segolodi[9], Refeletswe Lebelonyane[10], Molly Pretorius Holme[6], Sikhulile Moyo[7], Joseph Makhema[7], Shahin Lockman[6,7,11], Christophe Fraser[3], Myron Max Essex[6,7], Marc Lipsitch[1]*, On behalf of The Botswana Combination Prevention Project and PANGEA consortium

[1]Center for Communicable Disease Dynamics, Department of Epidemiology, Harvard T.H. Chan School of Public Health, Harvard University, Boston, United States; [2]Division of Molecular & Genomic Pathology, University of Pittsburgh Medical Center Presbyterian Shadyside, Philadelphia, United States; [3]Oxford Big Data Institute, Li Ka Shing Center for Health Information and Discovery, Nuffield Department of Medicine, Old Road Campus, University of Oxford, Oxford, United Kingdom; [4]Department of Biostatistics, Harvard T.H. Chan School of Public Health, Harvard University, Boston, United States; [5]Department of Statistics, The Wharton School, University of Pennsylvania, Philadelphia, United States; [6]Harvard T.H. Chan School of Public Health AIDS Initiative, Department of Immunology and Infectious Disease, Harvard T.H. Chan School of Public Health, Harvard University, Boston, United States; [7]Botswana Harvard AIDS Institute Partnership, Gaborone, Botswana; [8]Division of Global HIV/AIDS and TB, Centers for Disease Control and Prevention, Atlanta, United States; [9]HIV Prevention Research Unit, Centers for Disease Control and Prevention, Gaborone, Botswana; [10]Ministry of Health, Republic of Botswana, Gaborone, Botswana; [11]Brigham and Women's Hospital, Division of Infectious Diseases, Boston, United States

*For correspondence:
lmagosi@hsph.harvard.edu
(LEM);
mlipsitc@hsph.harvard.edu (ML)

## Abstract

**Background:** Mathematical models predict that community-wide access to HIV testing-and-treatment can rapidly and substantially reduce new HIV infections. Yet several large universal test-and-treat HIV prevention trials in high-prevalence epidemics demonstrated variable reduction in population-level incidence.

**Methods:** To elucidate patterns of HIV spread in universal test-and-treat trials, we quantified the contribution of geographic-location, gender, age, and randomized-HIV-intervention to HIV transmissions in the 30-community Ya Tsie trial in Botswana. We sequenced HIV viral whole genomes from 5114 trial participants among the 30 trial communities.

**Results:** Deep-sequence phylogenetic analysis revealed that most inferred HIV transmissions within the trial occurred within the same or between neighboring communities, and between similarly aged partners. Transmissions into intervention communities from control communities were more common than the reverse post-baseline (30% [12.2 – 56.7] vs. 3%

[0.1 – 27.3]) than at baseline (7% [1.5 – 25.3] vs. 5% [0.9 – 22.9]) compatible with a benefit from treatment-as-prevention.

**Conclusions:** Our findings suggest that population mobility patterns are fundamental to HIV transmission dynamics and to the impact of HIV control strategies.

**Funding:** This study was supported by the National Institute of General Medical Sciences (U54GM088558), the Fogarty International Center (FIC) of the U.S. National Institutes of Health (D43 TW009610), and the President's Emergency Plan for AIDS Relief through the Centers for Disease Control and Prevention (CDC) (Cooperative agreements U01 GH000447 and U2G GH001911).

## Editor's evaluation

The study by Magosi et al., evaluates the impact of targeted public health interventions on the HIV-1 transmission rate in Botswana. Using data from a large trial in Botswana, the authors found that HIV-1 transmission was more common to occur from control population groups into targeted population groups than vice-versa. The study is of public health interest, showing how some public health interventions are powerful in reducing HIV-1 transmission but only among the population targeted. This is a very comprehensive research study showing the advantages of using deep sequencing data in combination with phylogenetic tools to assess the positive impact of public health interventions in reducing HIV-1 transmission.

## Introduction

The global number of new infections with HIV-1, the virus that causes AIDS, has gradually declined since the peak in 1997, yet population-level HIV incidence remains high in East and Southern Africa, especially among young women (*UNAIDS, 2019*; *UNAIDS, 2021*). Despite efforts spanning four decades, there is still no successful HIV vaccine or widely administrable cure. This emphasizes the importance of curbing new infections in order to bring the HIV epidemic under control.

Four large cluster-randomized trials were recently conducted in East (Kenya and Uganda) and Southern Africa (Botswana, South Africa, and Zambia) to evaluate the effect of universal HIV testing and treatment in reducing population-level HIV incidence (*Hayes et al., 2019*; *Makhema et al., 2019*; *Havlir et al., 2019*; *Iwuji et al., 2018*). The trials showed variable outcomes, ranging from modest to no reduction in the occurrence of new HIV infections (*Abdool Karim, 2019*). The trials were motivated by evidence that early initiation of antiretroviral therapy substantially reduced onward transmission in people with HIV owing to suppressed virus levels (*Cohen et al., 2016*) and by mathematical models predicting that widely expanded access to HIV testing and treatment would rapidly and substantially reduce the occurrence of new infections (*Granich et al., 2009*).

The joint United Nations Programme on HIV/AIDS (UNAIDS) had set a global target of fewer than 500,000 new HIV infections by 2020 that was unmet. To stem the spread of new HIV infections we need to better understand patterns of HIV transmission in high-prevalence, generalized epidemics in sub-Saharan Africa. To obtain such understanding requires a better grasp of where and from whom those most at risk acquire infection and to whom they are likely to spread infection. Previous work in a high HIV burden setting in Kwa-Zulu Natal, South Africa identified age-disparate sexual partnerships between younger women under 25 years and older men as a primary contributor to HIV transmission and incidence in younger women (*de Oliveira et al., 2017*).

We hypothesized that geographic proximity, gender, age, and ready access to HIV care (randomized-HIV-intervention) contribute to shaping patterns of HIV transmission. To test these hypotheses, we performed a deep-sequence phylogenetic study of adults with HIV aged 16–64 years who consented to participate in the Botswana Ya Tsie HIV prevention trial to quantify the contribution of the above factors to HIV transmission patterns in Botswana adjusting for sampling heterogeneity.

Botswana is a sparsely populated, landlocked country in southern Africa, roughly the size of France, 581,730 km$^2$ (224,610 square miles), with a population of about 2.3 million people. The Kalahari Desert occupies approximately 70% of the country, and the population is largely distributed along an eastern corridor bordered by South Africa, Zambia, and Zimbabwe (*GISGeography, 2022*). It is a high middle-income country with a Gini index of 53.3 (*The World Bank, 2015*). One in five adults

in Botswana aged 15–49 years were reported to be living with HIV in 2019, representing the third-highest HIV prevalence in the world after Lesotho and Eswatini (*UNAIDS, 2020*).

Our results produced three key findings: first, most HIV transmissions inferred in the trial population occurred between partners residing in the same community or neighboring communities, and between partners of similar ages. Second, transmission events identified among trial participants sampled one or more years post-baseline were consistent with a greater flow of HIV transmissions into intervention communities from control communities than vice versa during the trial, although dates of transmission were not identified. Third, men and women sampled in the Botswana/Ya Tsie trial whose sexual partners could be inferred phylogenetically contributed similarly to the spread of HIV infection. Overall, our findings suggest that population mobility patterns are central to understanding HIV transmission dynamics and should be considered when designing and evaluating HIV control strategies.

## Materials and methods
### Overview of trial population
The Ya Tsie trial or Botswana Combination Prevention Project (BCPP) was a pair-matched community randomized trial to evaluate whether an optimized combination of effective HIV treatment and prevention interventions would substantially reduce population-level HIV incidence over 29 months. The trial was conducted from 2013 to 2018 in 30 rural and peri-urban communities (average population size: 5855) in the Central, South-East, and North/North-East regions of Botswana that represented an estimated total population of 175,664 (7.6% of the national population) (*Figure 1*). Among eligible community residents, 13,131 adults aged 16–64 years gave informed consent to participate in the trial. See *Makhema et al., 2019*; *Gaolathe et al., 2016* for detailed eligibility criteria. The estimated HIV prevalence among participating communities at the start of the trial was 25.7% (27,446 / 106,712) (*Supplementary file 1-Table 1A*) (see Materials and methods section on computing age-gender estimates of the number of people with HIV in each trial community). Communities were matched into 15 pairs according to population size and age structure, access to health services, and geographic location relative to major urban centers; communities in each pair were then randomized into intervention and control arms. Intervention communities received expanded HIV testing, early initiation of antiretroviral treatment, and strengthened linkage-to-care, for example, clinic referrals with appointment dates, text alert reminders of appointments, and following-up with those who missed appointments. Intervention communities also received wider access to prevention of mother-to-child transmission services and male circumcision services, compared with control communities that received the standard of care.

### Data set of people with HIV in the Botswana/Ya Tsie trial
To better understand HIV transmission patterns among participating communities, residents with HIV-1 were invited to provide a sample for viral phylogenetic analyses. This included (1) all people with HIV-1 identified during a baseline survey of 20% of households randomly sampled from each trial community to establish an incidence follow-up cohort, (2) all people with HIV-1 identified through annual household surveys conducted in this 20% household sample in all 30 communities during the trial, (3) all people with HIV-1 who were enrolled in a community-wide survey in 100% of 6 communities, 3 in the intervention and 3 in the control arms (the 'end of study survey'), (4) all newly identified people with HIV-1 discovered during an initial community-wide HIV testing and counseling campaign followed by more targeted HIV testing in the 15 intervention communities, (5) all people with HIV-1 in intervention communities who later presented at healthcare facilities after being identified during community-wide testing campaigns, and (6) all people with HIV-1 who were already receiving HIV care at health facilities in the 15 intervention communities. Altogether, 5114 participants consented to a blood draw for HIV-1 viral genotyping and their viral genomes were successfully deep-sequenced, representing the largest phylogenetics study conducted in Botswana (*Supplementary file 1-Table 1A*).

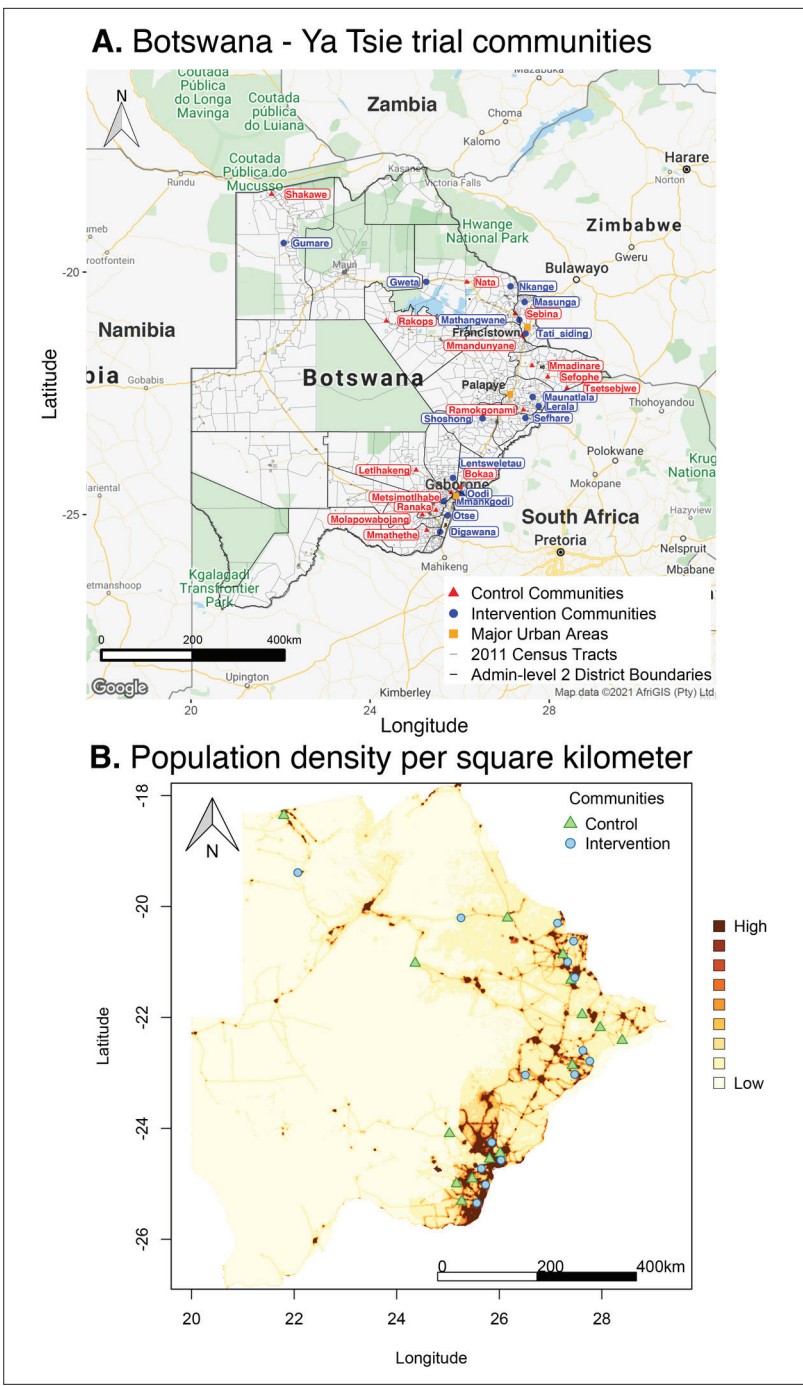

**Figure 1.** Location of 30 rural and peri-urban communities that participated in the Botswana/Ya Tsie trial. (**A**) A map of Botswana showing the spatial distribution of intervention and control communities in the Botswana/Ya Tsie trial. Intervention communities are denoted by filled blue circles and control communities are represented by filled red triangles. Trial communities are shown in the context of level-2 administrative subdivisions (solid black lines) and 2011 census enumeration areas (solid gray lines). Major urban areas are symbolized as filled orange squares. (**B**) Spatial distribution of the estimated population density of persons per square kilometer in Botswana in 2011. Filled blue circles represent the location of intervention communities and filled green triangles denote control communities in the Botswana/Ya Tsie trial.

The online version of this article includes the following figure supplement(s) for figure 1:

*Figure 1 continued on next page*

*Figure 1 continued*

**Figure supplement 1.** Characterization of deep-sequenced HIV-1 viral whole genomes of 4246 trial participants (88% had viral genomes sequenced from pro-viral DNA) sequenced at the Harvard biopolymers sequencing core facility.

**Figure supplement 2.** Characterization of deep-sequenced HIV-1 viral whole genomes of 868 trial participants (45% had viral genomes sequenced from pro-viral DNA) sequenced at the Wellcome Trust Sanger Institute.

## Computing age-gender estimates of the number of people with HIV in each trial community

### Age-gender HIV prevalence estimates in each trial community

To obtain HIV prevalence estimates we first fit a random-intercept Poisson regression model of HIV status, with age and gender as fixed-effects, using nonparametric maximum likelihood estimation in the gllamm program (*Sophia, 1999*; *Rabe-Hesketh and Skrondal, 2012*) in Stata 13.1. The model was fit to data indicating the HIV status of 12,570 participants that had consented to an HIV test and were part of a baseline survey of 20% of households randomly sampled from each of the 30 trial communities to establish an incidence follow-up cohort; the fraction of positive tests for HIV-1 infection was, 28.6% (3596 / 12,570). Ages of participants were grouped in two ways, as 5-year age-categories and four age-categories: 16–24, 25–34, 35–49, and 50–64 years. Furthermore, the counts of people with HIV for a specific age-gender grouping, $i$ in community, $j$ were assumed to have a Poisson distribution with mean, $u_{ij}$ so that

$$\ln\left(u_{ij}\right) = \ln\left(e_{ij}\right) + \beta_1 + \beta_2 x_{2ij} + \beta_3 x_{3ij} + \zeta_j \ . \tag{1}$$

Accordingly, $\beta_1$ denotes the intercept, $\zeta_j$ the predicted random-intercept representing unobserved heterogeneity across trial communities, $\beta_2 x_{2ij}$ and $\beta_3 x_{3ij}$ fixed-effect covariates for age group and gender, respectively, and $\ln\left(e_{ij}\right)$ an offset where $e_{ij}$ represents the number of people tested for HIV in a specific age-gender grouping, $i$ in community, $j$ (i.e., exposed) so that,

$$\ln\left(u_{ij}\right) - \ln\left(e_{ij}\right) = \beta_1 + \beta_2 x_{2ij} + \beta_3 x_{3ij} + \zeta_j. \tag{2}$$

### Computing empirical Bayes estimates of HIV prevalence

Next, to account for heterogeneity in HIV prevalence across trial communities we computed empirical Bayes estimates (or posterior means) for the age-gender specific HIV prevalence in each community. The empirical Bayes estimate of HIV prevalence for the *jth* community-age-gender combination is

$$\hat{\zeta}_j EB = \hat{R}_j \hat{\zeta}_j ML \tag{3}$$

where $\hat{R}_j = \frac{\hat{\psi}_j}{\hat{\psi}_j + \frac{\hat{\theta}}{\hat{n}_j}}$ .

Here, $\hat{R}_j$ denotes a shrinkage factor that describes the proportion of the total variation, $\hat{\psi}_j + \frac{\hat{\theta}}{\hat{n}_J}$ attributed to differences in the age-gender specific HIV prevalence among trial communities, $\hat{\psi}_j$ . Furthermore, $\hat{\zeta}_j ML$ specifies the maximum likelihood estimate of $\zeta_j$ . Empirical Bayes estimates were obtained with the gllapred program (*Sophia, 1999*; *Rabe-Hesketh and Skrondal, 2012*) in Stata 13.1.

### Estimates of size of population with HIV in trial communities

We thereafter estimated the size of the population with HIV for each age-gender category in each trial community. Empirical Bayes estimates of the age-gender specific HIV prevalence for each trial community were standardized to the proportion of participants in the 2011 Botswana population and housing census that were in the same age and gender categories of these communities. Then for a specific community-age-gender combination, for example, females aged 16–24 years in community $j$, the estimated number of people with HIV, $\hat{N}_{HIV+[Female, \ 16-24]}$ is computed as

$$\hat{N}_{HIV+[Female, \ 16-24]} = \hat{N}_{total} * E * \{Pr_{census}[Female, 16 - 24 \ years] *$$
$$Pr[HIV+ \mid Female, \ 16 - 24 \ years]\} \tag{4}$$

where.

$\hat{N}_{total}$: Estimated population size for the jth trial community based on plot and household enumeration data (**Makhema et al., 2019**).

$E$: Proportion of enumerated household members that were eligible to participate in the trial.

$\text{Pr}_{census}$ [Female, $16 - 24$ years]: Joint probability denoting the proportion of census participants who are female and aged 16–24 years within community j.

$\text{Pr}[HIV+ \mid Female, 16 - 24 \, years]$: Empirical Bayes estimate of HIV prevalence for females aged 16–24 years that participated in a baseline survey of 20% of households randomly sampled from community j.

The age-gender estimates of the number of people with HIV in each trial community were used to adjust for sampling heterogeneity when estimating the proportions of HIV transmissions in the trial population (see Materials and methods section on adjustment for variable sampling rates across different demographic groups or randomized-HIV-interventions). Population-size estimates of people with HIV were computed in R version 3.5.2 (**R Deveploment Core Team, 2020**).

## Deep-sequencing of HIV viral genomes for phylogenetic analyses

Because most participants were virally suppressed at the time of sampling, deep sequencing of HIV-1 viral genomes was generally done on proviral DNA (81%, n = 4142) (**Novitsky et al., 2015**) (see Materials and methods section on paired-end deep-sequencing of HIV viral genomes for phylogenetic analyses). We deep-sequenced near-full length viral genomes to optimize resolution and power to detect viral genetic clusters of similar HIV-1 infections (**Dennis et al., 2014**; **Figure 1—figure supplements 1 and 2**). For each participant, we thus obtained mapped short reads generated with deep-sequencing that capture the HIV-1 viral diversity in an individual and a corresponding consensus sequence that represents a summary of the HIV-1 viral population in that individual. Consensus sequences were generated from alignments of the mapped short reads by identifying the majority nucleotide call at each base position along the HIV-1 genome (**Wymant et al., 2018a**).

## Paired-end deep-sequencing of HIV viral genomes for phylogenetic analyses

Paired-end deep-sequencing was performed with Illumina MiSeq and HiSeq instruments at two locations, the BioPolymers sequencing core facility at Harvard Medical School, Boston, United States (n = 4246 participants) (**Figure 1—figure supplement 1**), and at the Wellcome Trust Sanger Institute, Hinxton, United Kingdom (n = 868 participants) (**Figure 1—figure supplement 2**) through the PANGEA consortium (**Pillay et al., 2015**). The sequencing success rate was relatively high, 75%–80% for first round amplicons and 95%–98% for second round amplicons. Moreover, the quality of sequencing was assessed with standard metrics for deep (or next-generation) sequencing data; however, we cannot exclude the potential for sequencing errors arising from hypermutations. The shiver sequence assembly software (**Wymant et al., 2018a**) was used to assemble and map each participant's deep-sequencing short reads to a de-novo reference sequence tailored to the participant's viral population. A listing of command-line parameters used to assemble HIV viral whole genomes with Shiver is provided in **Supplementary file 1-Table 1B**.

## Criteria for inclusion in phylogenetic analyses

Our analysis was restricted to samples from 3832 participants (approximately 14% of the estimated 27,446 individuals living with HIV in trial communities) who had available demographic information and met minimum criteria on genome missingness (≥6300 nucleotides available) for inclusion in phylogenetic analyses. More precisely, individuals whose viral consensus genomes had fewer than 30% of the bases missing beyond the first 1000 nucleotides were retained for analysis. Sampling (or genotyping) density is defined as the estimated proportion of individuals living with HIV in a trial community that were included in our sample. Intervention communities were sampled more densely, 16% (2281 / 14,263) than control communities, 12% (1551 / 13,183); and sampling densities among all 30 trial communities ranged from 2.7% to 36.2% (**Figure 2**, **Supplementary file 1**, Table 1C) (see Materials and methods section on computing age-gender estimates of the number of people with HIV in each trial community). Based on the level of sexual mixing predicted for the trial (**Novitsky et al., 2013**)—that 21% (standard error: 2.6%) of the relationships would be out-of-community partnerships (assumed for this calculation to be with communities that were not in the trial)—and a genotyping

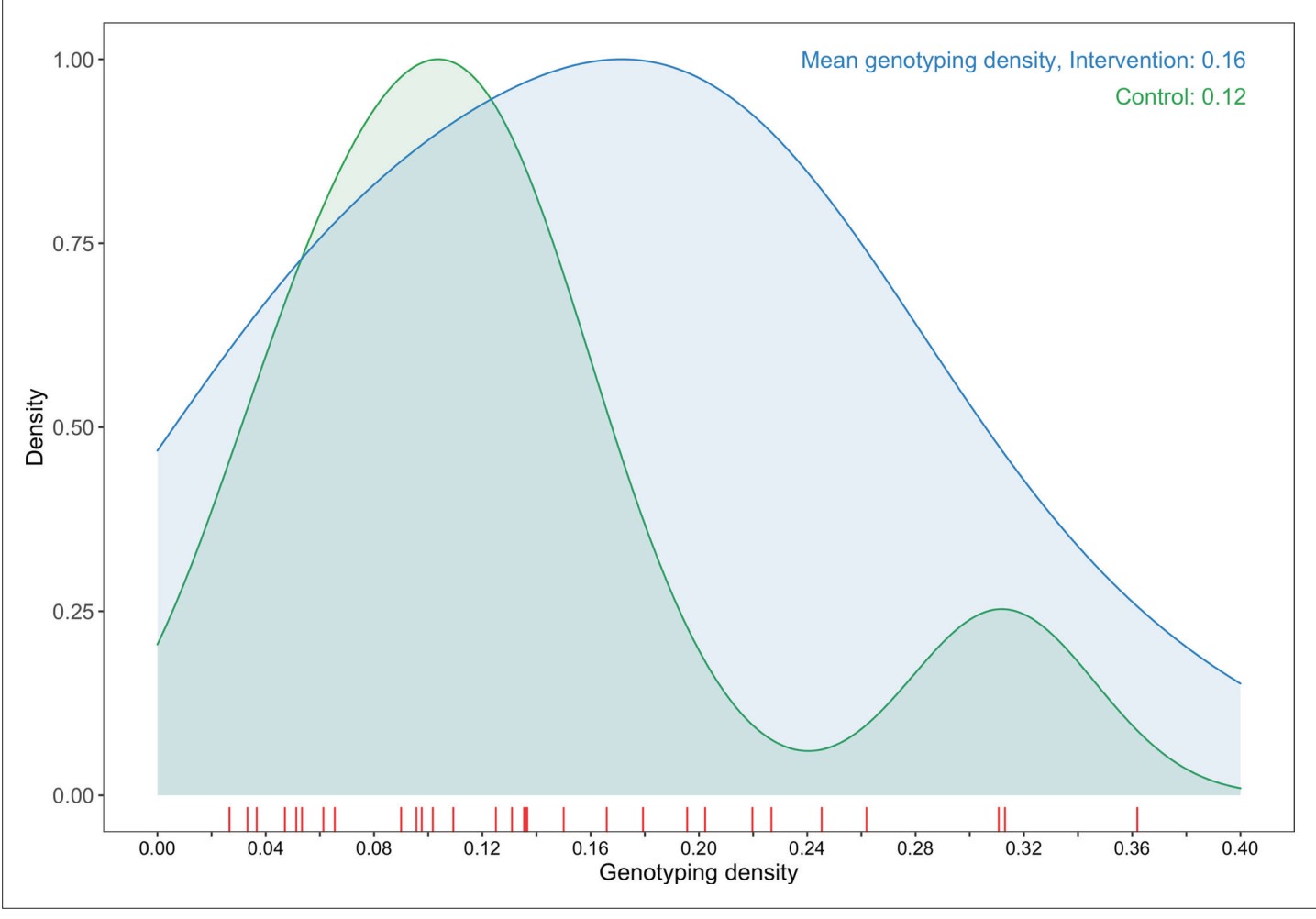

**Figure 2.** A plot comparing sampling densities of HIV-1 viral genomes in intervention and control communities in the Botswana/Ya Tsie trial. Intervention communities (blue curve) were sampled more densely than control communities (green curve). Raw data points for the density plots are displayed as a rug-plot (red) on the x-axis. The overall genotyping density across all 30 trial communities was 14% (3832 / 27,446).

density of 14% we expected our data set to contain the viral transmitter to a genotyped individual in about one out of every nine cases of HIV-1 transmission, that is, 0.14 * (1 − 0.21) = 0.11. This estimate reflects simplifying assumptions that: (i) for a pair of individuals whose viral sequences are genetically linked, the probability of sampling one member of the pair is independent from that of sampling the other member of the pair such that the probability of observing a pair that is genetically linked is the product of the individual sampling probabilities and (ii) the sampling probabilities were equal for men and women with HIV. The genotyping densities for women and men were 15% (2730 / 17,815) and 11% (1102 / 9631), respectively. Therefore, we expected the proportion of female viral transmitters in our data set to be 0.153 * (1 − 0.21) = 0.12 and male viral transmitters to be 0.114 * (1 − 0.21) = 0.09.

## Definition of seroconverters

The HIV-incidence prospective cohort of the Botswana/Ya Tsie trial comprised a random sample of 20% of households in each community to evaluate the incidence of HIV infection and adoption of the intervention during the trial. The 3832 participants included in phylogenetic analyses included 85% (124 / 146) of the seroconverters in the HIV-incidence cohort; seroconverters in the HIV-incidence cohort were defined as individuals that acquired HIV infection during the trial (**Supplementary file 1-Table 1C**). Seven of the 2465 baseline samples included in phylogenetic analyses were seroconverters in the HIV-incidence cohort (i.e., individuals with a negative-HIV test and a subsequent positive-HIV test

sample over the course of less than a year, and thus defined as baseline) compared with 117 of the 1367 post-baseline samples.

## Comparing genetic distances between HIV-1 viral consensus sequences of trial participants included in phylogenetic analyses

We compared the distances between all possible pairs (n=7,340,196) of the 3832 HIV-1 viral whole-genome consensus sequences available for phylogenetic analyses (*Figure 3A*). Sequences were codon-aligned to HXB2, a standard HIV reference, with MAFFT v7.407 (*Katoh et al., 2002*; *Katoh and Standley, 2013*) and HIVAlign (https://www.hiv.lanl.gov/content/sequence/VIRALIGN/viralign.html); genetic distances were then computed under the Tamura-Nei-1993 nucleotide substitution model (*Tamura and Nei, 1993*) with TN93 v1.0.6 (https://github.com/veg/tn93 *iGEM/UCSD evolutionary biology and bioinformatics group, 2021*). A listing of parameters used to align sequences with MAFFT and HIVAlign, and compute genetic distances with TN93 is provided in *Supplementary file 1-Table 1B*.

## Consensus sequence phylogenetics to identify clusters of participants with genetically similar HIV-1 infections

To save time and computational resources from evaluating sequences that were too distantly related, we first identified clusters of participants with genetically similar HIV-1 infections as a filtering step, before performing ancestral host state reconstruction with Phyloscanner to detect probable directed transmission pairs. Two clustering algorithms, HIV Transmission Cluster Engine (HIV-TRACE) v0.4.4 (*Wertheim et al., 2014*; *Kosakovsky Pond et al., 2018*) and Cluster Picker v1.2.3 (*Ragonnet-Cronin et al., 2013*), were used to identify clusters of individuals whose HIV-1 viral whole-genome consensus sequences were genetically similar—suggesting they were probably members of a transmission chain (*Poon, 2016*; *Rose et al., 2017*). HIV-TRACE defines clusters based on pairwise genetic distances only; comparatively, Cluster Picker identifies clusters using pairwise genetic distances with the guidance of a phylogenetic tree. A multiple sequence alignment (as described in the Materials and methods section on comparing genetic distances between HIV-1 viral consensus sequences of trial participants included in phylogenetic analyses) was provided as input to HIV-TRACE and Cluster Picker. Additionally, for cluster picker, a corresponding phylogenetic tree inferred with FastTree2 v2.1.10 and boot-strap support values approximated with the Shimodaira-Hasegawa test (*Shimodaira and Hasegawa, 1999*) were provided as inputs. We defined genetic similarity clusters as groups of two or more participants whose viral whole-genome consensus sequences were separated by a genetic distance at or smaller than 4.5% nucleotide substitutions per site—and, for Cluster Picker, a bootstrap support value of at least 80%. The genetic distance threshold of 4.5% nucleotide substitutions per site was motivated by the distribution of genetic distances separating HIV-1 subtype C viral whole genomes of epidemiologically linked couples in the HIV Prevention Trials Network (HPTN) 052 trial (*Figure 3B*; *Cohen et al., 2016*; *Eshleman et al., 2011*). A listing of parameters used for consensus-sequence phylogenetics with HIV-TRACE and Cluster Picker is provided in *Supplementary file 1-Table 1B*.

## Deep-sequence phylogenetics to infer the probable order of transmission events within identified clusters of genetically similar HIV-1 infections

We performed parsimony-based ancestral host state reconstruction of the mapped deep-sequencing short reads of participants in clusters of genetically similar HIV-1 infections (n = 525) with the Phyloscanner software v1.8.0 (*Grabowski et al., 2018*; *Ratmann et al., 2019*; *Wymant et al., 2018b*) to identify probable transmission pairs within clusters and the probable direction of transmission between them. Phyloscanner identifies potential transmission pairs by inferring whether the viral population in an individual is ancestral to or descendent from that of another individual, and crucially addresses contamination by excluding duplicates and phylogenetic outliers. To infer phylogenetic linkage and direction of transmission for a set of individuals Phyloscanner aligns submitted mapped reads in sliding windows along the genome that are matched across individuals; and at each window, infers a phylogeny for ancestral state reconstruction to identify probable ancestral relationships between all possible pairs of individuals in the set. Pairs of individuals are then classified as phylogenetically linked or unlinked based on the distance, adjacency, and topology of their subgraphs. A subgraph

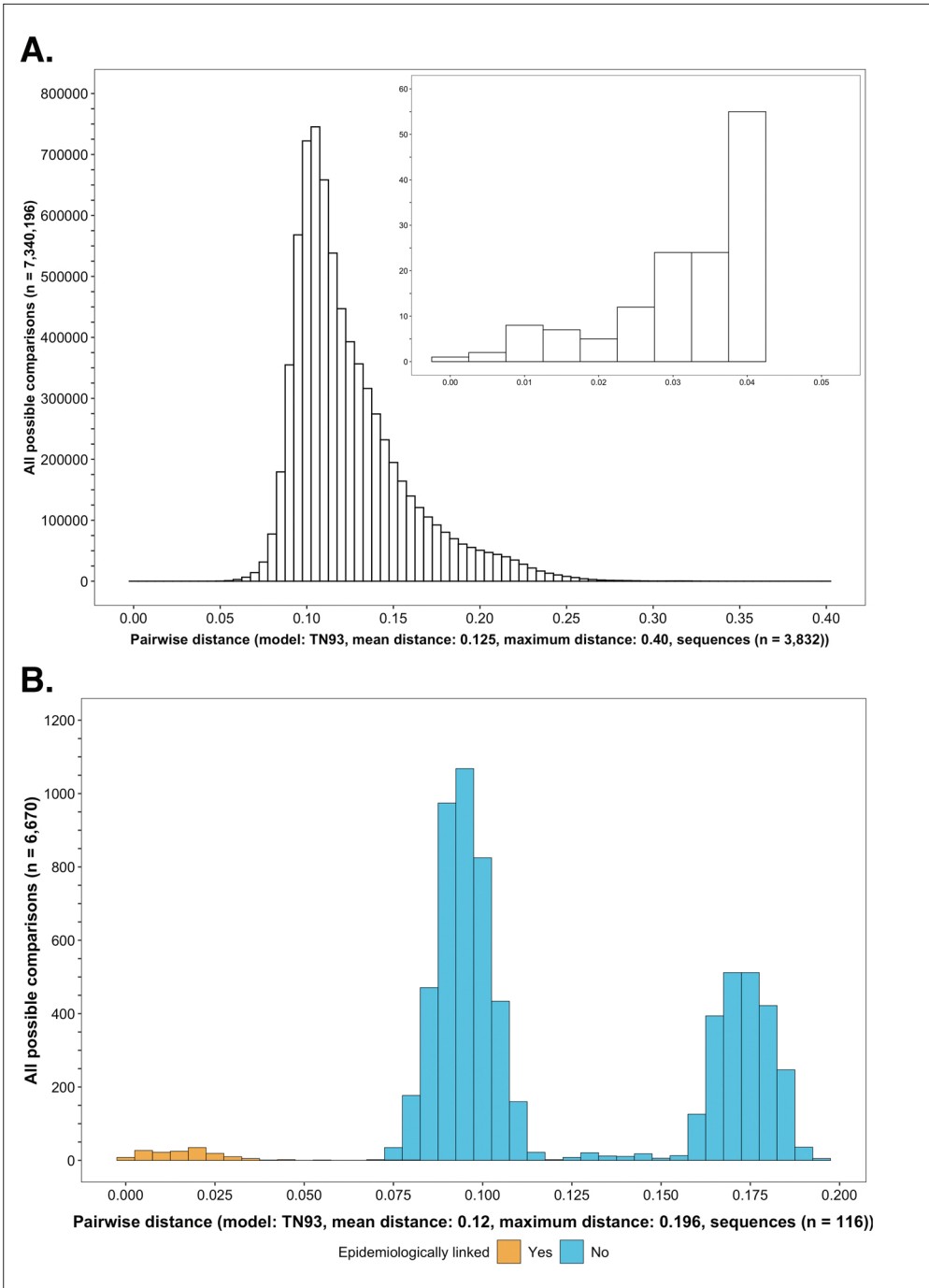

**Figure 3.** Histograms showing the distribution of genetic distances between HIV-1 viral whole-genome consensus sequences. (**A**) A histogram showing the distribution of genetic distances between HIV-1 viral whole-genome consensus sequences of trial participants included in phylogenetic analyses. Out of the 5114 trial participants that consented to viral genotyping 3832 met criteria for inclusion in phylogenetic analyses; these were individuals whose viral genomes had fewer than 30% of the bases missing beyond the first 1000 nucleotides and a minimum genome length of 6300 nucleotides. Tamura-Nei 93 genetic distances were computed between HIV-1 viral whole-genome consensus sequences of all possible pairs (n = 7,340,196) of the 3832 trial participants available for phylogenetic analyses. To improve visibility pairwise genetic distances below 0.05 (5%) nucleotide substitutions per site are also shown in a zoomed-in plot. The mean (± standard deviation) pairwise distance separating sequences was 12.5 ± 3.0%, and the maximum pairwise distance was 40.0% substitutions per site. (**B**) A histogram showing the distribution of genetic distances between viral whole-genome consensus sequences of epidemiologically linked HIV-1 subtype-C couples in the HIV Prevention Trials Network 052 study. Tamura-Nei 93 genetic distances

*Figure 3 continued on next page*

*Figure 3 continued*

were computed between viral whole-genome consensus sequences of all possible pairs (n = 6670) of the 116 epidemiologically linked couples with HIV-1 subtype-C infections in the HIV Prevention Trials Network 052 study. Pairwise comparisons between epidemiologically linked sequences (yellow) and ones between epidemiologically unlinked sequences (blue) are highlighted in color. The mean (± standard deviation) pairwise distance separating the sequences was 12.0 ± 4.1%, and the maximum pairwise distance was 19.6% substitutions per site. Genetic distances separating viral sequences of most epidemiology-linked couples were below 5.0% substitutions per site.

in Phyloscanner refers to all tips and internal nodes of a phylogeny assigned to an individual through parsimony-based ancestral state reconstruction. A pair of individuals, *i* and *j* are considered to be phylogenetically linked, more specifically, phylogenetic linkage is not excluded when the minimum patristic distance between their subgraphs, $\Delta_{ij}$ is below a set threshold, and *i* and *j* are adjacent meaning that the shortest path connecting their subgraphs in a phylogeny does not pass through a third individual. Tree topology is used to infer the probable direction of transmission between a pair of individuals and refers to the number of subgraphs belonging to each individual in the pair that are ancestral-to or descendant-from those of the other individual. We classified pairs of individuals separated by a subgraph distance below 0.035 substitutions per site as phylogenetically linked based on previous work by *Ratmann et al., 2019*. The subgraph distance distribution of all pairs identified with Phyloscanner within genetic similarity clusters is shown in *Figure 4*. A listing of command-line parameters used for deep-sequence phylogenetics with Phyloscanner is provided in *Supplementary file 1-Table 1B*. Identified genetic similarity clusters were analyzed with Phyloscanner in parallel for computational efficiency using shell scripts that are available upon request.

## Identifying probable source-recipient pairs with strong phylogenetic evidence for linkage and direction of transmission

Probable transmission pairs identified with Phyloscanner were further classified on the strength of evidence for phylogenetic linkage, $\hat{\lambda}_{ij}$ to separate pairs with strong support for linkage and direction of transmission from other potentially linked pairs; accounting for the extent of overlap in the read alignments from which the deep-sequence phylogenies were inferred. The strength of phylogenetic evidence or linkage score is described as, $\hat{\lambda}_{ij} = \frac{k_L}{n}$ where *n* is the number of deep-sequence phylogenies inferred in windows along the genome for which individuals *i* and *j* had sufficient deep-sequence reads for phylogenetic inference; and $k_L$ is the number of deep-sequence phylogenies along the genome that support a specific phylogenetic relationship type between individuals *i* and *j*, for example, that the subgraphs of individual *i* are ancestral-to, descendent-from, sibling-to, or intermingled-with those of individual *j*; sibling and intermingled topologies are ones where there is insufficient evidence to make inferences about the probable direction of transmission (*Ratmann et al., 2019*). We used a linkage and direction of transmission score threshold of 57% $\left(\hat{\lambda}_{ij} > c;\ c = 0.57\right)$ or on average at least 24 out of 42 windows supporting linkage and direction of transmission for a highly supported transmission pair. Our Phyloscanner analyses were based on windows or mapped read alignments that were 200-bp long, affording *n* = 42 (8400 / 200) non-overlapping alignments and deep-sequence trees along the genome; we excluded the first 1000 nucleotides of the HIV-1 genome from analysis as they are typically poorly sequenced corresponding to a genome length of 8400 bp. The linkage and direction of transmission score threshold, $c \in (0, 1)$ was selected such that the posterior probability for $\hat{\lambda}_{ij} > 50\%$, at least half of the windows along the genome supporting an ancestral relationship type between a pair of individuals, *i* and *j* exceed 80%, $p\left(\hat{\lambda}_{ij} > 0.5 \mid k_L,\ n\right) > \alpha$, where $\alpha = 80\%$ and $k_L \sim \text{Binomial}\left(n,\ \hat{\lambda}_{ij}\right)$. The linkage and direction of transmission score threshold were computed with the Bayesian binomial test function in the BayesianFirstAid R package v0.1.

## Estimating error rates in phylogenetic inference of direct transmission between sampled males and females

HIV-1 infections in southern Africa are more commonly transmitted through heterosexual contact between males and females, compared with sexual contact between same-sex couples. Thus, we can use the probability of inferring a phylogenetically linked male-male pair to calibrate an upper bound on the false discovery rate (FDR) of inferring a phylogenetically linked male-female pair, if we assume

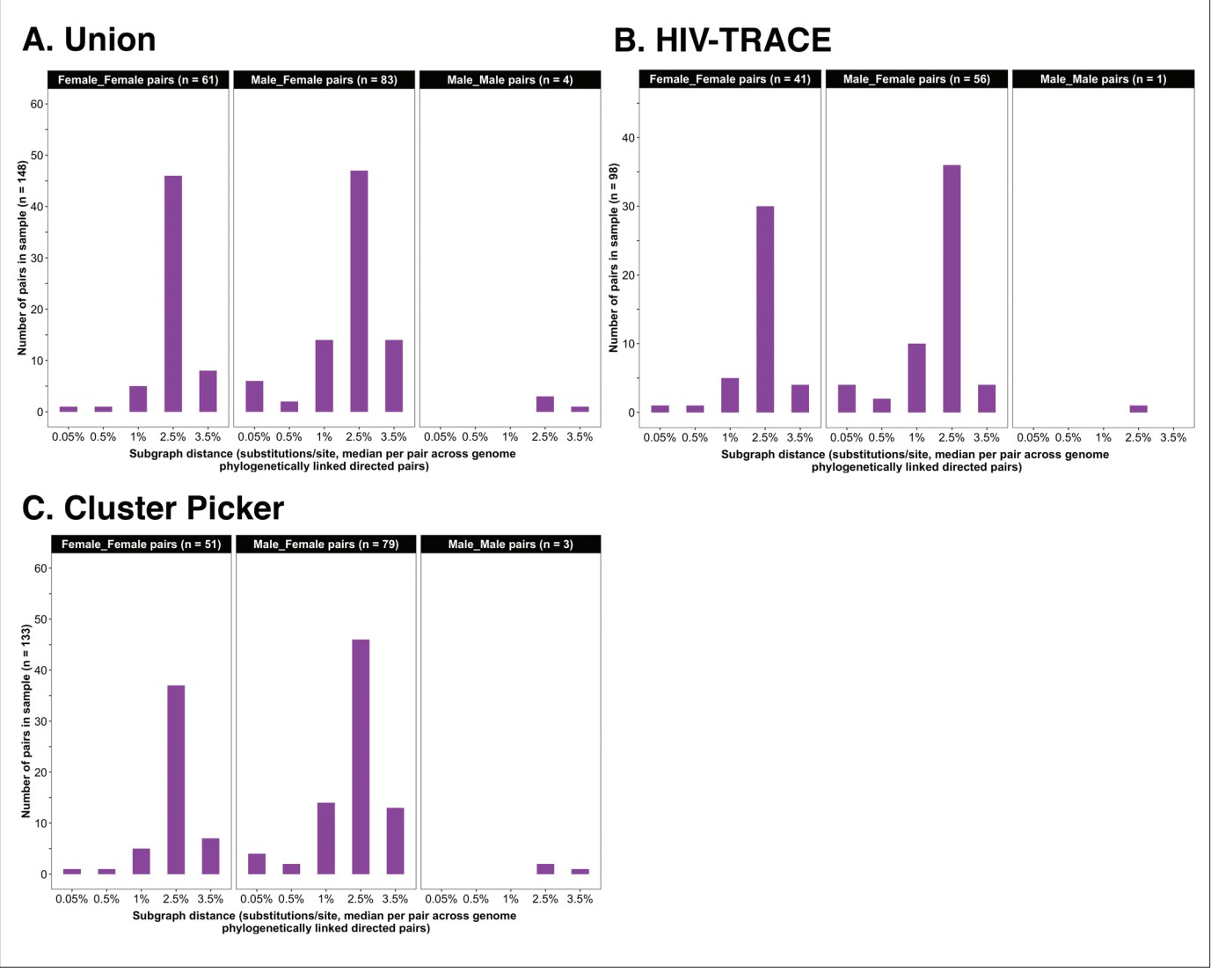

**Figure 4.** Barplots showing subgraph distance distributions between deep-sequenced HIV-1 viral whole genomes of trial participants in clusters of genetically similar HIV-1 infections. Within each genetic cluster parsimony-based ancestral host state reconstruction was done with Phyloscanner (*Grabowski et al., 2018*; *Ratmann et al., 2019*; *Wymant et al., 2018b*) to identify probable transmission pairs and resolve the probable order of transmission events. Thereafter, within each identified genetic cluster, the median subgraph distance and most frequent subgraph topology between each pair of individuals were determined across all deep-sequence phylogenies along the genome where the pair had sufficient mapped reads for phylogenetic inference. A subgraph refers to all tips and internal nodes of a phylogeny assigned to an individual through parsimony-based ancestral state reconstruction. Subgraph distances were standardized to the mean rate of evolution for HIV-1 group-specific antigen (*GAG*) and polymerase (*POL*) genes to account for variation in mutation rates along the genome. The three panels show median subgraph distance distributions of pairs in the union (**A**) of genetic clusters detected with HIV-TRACE (**B**) and Cluster Picker (**C**) algorithms. Within each panel, phylogenetically linked pairs with an inferred direction of transmission are summarized into same-sex (male-male and female-female) and opposite-sex (male-female) pairs. The subgraphs of deep-sequenced HIV-1 viral genomes of most phylogenetically linked directed pairs were separated by a patristic distance below 0.035 (3.5%). Same-sex pairs were presumed to be members of transmission chains with missing intermediates.

the probability of erroneously inferring direct transmission between a male-female pair is the same as the probability of inferring a phylogenetically linked male-male pair (*Ratmann et al., 2019*).

## Calibrating a false discovery rate for inferring direct transmission

Consider a deep-sequence phylogenetic analysis where $L_{mm}$ male-male pairs and $L_{mf}$ male-female pairs were phylogenetically inferred in the sequenced sample. For $S_m$ successfully sequenced males

and $S_f$ successfully sequenced females, let the probability of inferring a phylogenetically linked male-male pair in the sample be

$$\frac{L_{mm}}{S_m * (S_m - 1)/2} \, ,\tag{5}$$

where $L_{mm}$ denotes the number of phylogenetically linked male-male pairs that were identified and $S_m * (S_m - 1)/2$ represents the number of distinct possible male-male pairs that could have been identified in the sample. The estimated number of phylogenetically linked male-female pairs between whom direct transmission did not occur would then be

$$\hat{F}^c_{mf} = \frac{L_{mm}}{S_m * (S_m - 1)/2} * S_m * S_f \, .\tag{6}$$

Thereby affording a FDR estimate of,

$$\hat{\rho}^c_{mf} = \frac{\hat{F}^c_{mf}}{L_{mf}}\tag{7}$$

This represents an upper bound on the FDR of inferring direct transmission between males and females in phylogenetically linked male-female pairs.

## Adjustment for variable sampling rates across different demographic groups or randomized-HIV-interventions

To estimate HIV transmission flows in the trial population within and between different age-gender groups and locations (trial communities, trial arms, and geographic regions), accounting for variable rates of sampling, we used the method described by *Carnegie et al., 2014*. The method by Carnegie et al. uses a frequentist approach, alternatively a Bayesian approach could be employed as described by *Ratmann et al., 2020*.

### Data set and sampling probability

Consider a population of individuals with HIV, $\Omega$ of finite size, $N$ partitioned into $u = 1, \cdots, G$ disjoint groups so that, $N_u$ represents the number of individuals with HIV in group, $u$. Groups might represent communities (or geographic locations), age-categories, genders, or trial arms. Let us suppose that we would like to estimate the proportions of HIV transmissions that occurred within and between intervention and control communities in the Botswana/Ya Tsie trial so that group, $u$ are individuals residing in intervention communities and group, $v$ are individuals residing in control communities. We denote $n_u$ and $n_v$ as the number of individuals randomly sampled from groups, $u$ and $v$, respectively, whose HIV-1 viral genomes were successfully sequenced; yielding a sampling probability, $s_u = n_u / N_u$ for group, $u$ and similarly, $s_v = n_v / N_v$ , for group $v$. The vector

$$z = \left( z_{uu}, \ z_{uv}, \ z_{vu}, \ z_{vv} \right)\tag{8}$$

describes the counts of directed (source-recipient) transmission pairs identified in the sequenced sample within and between intervention and control communities, where $z_{uu}$ denotes transmission pairs within intervention communities; $z_{vv}$ denotes transmission pairs within control communities; $z_{uv}$ denotes transmission pairs for whom a member of an intervention community was the inferred source of transmission to a control community member and vice-versa, $z_{vu}$ . We assume that HIV transmission pairs that are absent from the sequenced sample are missing at random conditional on group membership; meaning that the identified probable source-recipient transmission pairs are a random sample of the source-recipient transmission pairs present in the trial population. Let

$$\mathrm{n}_{uv} = \begin{cases} n_u \ * \ n_v, & u \neq v \\ n_u \left( n_u - 1 \right) /2, & u = v \end{cases}\tag{9}$$

be the number of distinct possible transmission pairs among individuals sampled from $u$ and $v$; similarly, let $N_{uv} = N_u * N_v \left( \text{or } N_u \left( N_u - 1 \right) /2 \ \text{if } u = v \right)$ be the number of distinct transmission pairs available between trial population groups, $u$ and $v$.

## Probability of viral linkage between a pair of sampled individuals

Estimates of probabilities of linkage between two individuals randomly sampled from within demographic groups are observed using the method of *Carnegie et al., 2014*. The method assumes that samples are missing at random from within demographic groups and that relative increase in weight of individuals from a given group between the observed and true sample is the same as relative increase in weight between sub-samples of the observed data—obtained from the observed sampling rate—and the observed data.

In accordance with *Carnegie et al., 2014* and accounting for the probable direction of transmission, we estimate the probability of viral linkage between two individuals randomly sampled from their respective groups for each group-pairing as

$$\hat{p} = \left( \hat{p}_{uu}, \hat{p}_{uv}, \hat{p}_{vu}, \hat{p}_{vv} \right) \tag{10}$$

where, $\hat{p}_{ij} = \frac{z_{ij}}{n_{ij}}$, $i = u, v$ and $j = u, v$.

## Proportion of HIV transmissions in the trial population adjusted for sampling heterogeneity

We next estimate the relative probability that an HIV transmission event in the trial population occurred from sexual contact between partners residing within intervention communities or within control communities; compared with partners where one resides in an intervention community and the other a control community or vice versa. Accordingly, we estimate the conditional probability, $\hat{\theta}_{ij}$ that a pair of individuals are from a specific group-pairing given that their HIV-1 viral genomes are linked as,

$$
\begin{aligned}
\hat{\theta}_{ij} &= P \left( \text{pair is from groups, } i \text{ and } j \mid \text{pair is linked} \right) \\
&= \frac{N_{ij}\hat{p}_{ij}}{\sum_m \sum_{n \geq m} N_{mn}\hat{p}_{mn}}, i = u, v \text{ and } j = u, v.
\end{aligned}
\tag{11}
$$

Here, $N_{ij}\hat{p}_{ij}$ denotes the estimated number of HIV transmissions in the trial population attributed to groups, $i$ and $j$. Similarly, $\sum_m \sum_{n \geq m} N_{mn}\hat{p}_{mn}$ represents the number of HIV transmissions in the trial population for all groups-pairings. Thus, the vector

$$\hat{\theta} = \left( \hat{\theta}_{uu}, \ \hat{\theta}_{uv}, \ \hat{\theta}_{vu}, \ \hat{\theta}_{vv} \right) \tag{12}$$

describes estimated proportions of HIV transmissions in the trial population within and between intervention and control communities adjusted for differential sampling by trial arm.

## Computation of confidence intervals for estimated HIV transmission flows in the trial population

We use the Goodman method (*Goodman, 1965*) to compute simultaneous confidence intervals (CIs) at the 5% significance level for the estimated proportions of HIV transmissions within and between population groups (or strata). A continuity correction factor is implemented for the Goodman method's CIs to account for small sample sizes (*Cherry, 1996*). The Goodman method estimates simultaneous CIs for the parameters of a multinomial distribution and thus assumes independence of observations.

To estimate Goodman's CIs the estimated number of HIV transmission pairs in the trial population within and between intervention and control communities are scaled such that the sum of the weights is equal to the total number of probable male-female transmission pairs identified in the deep-sequenced sample (n = 82), and the weights are treated as known; broadly similar to calibration of weights in survey sampling. For example, the weighted counts of probable male-female transmission pairs within and between intervention and control communities are given by the vector,

$$
\begin{aligned}
\delta &= z^+ * \left( \hat{\theta}_{uu}, \ \hat{\theta}_{uv}, \ \hat{\theta}_{vu}, \ \hat{\theta}_{vv} \right) , \\
&= \left( z^+\hat{\theta}_{uu}, \ z^+\hat{\theta}_{uv}, \ z^+\hat{\theta}_{vu}, \ z^+\hat{\theta}_{vv} \right)
\end{aligned}
\tag{13}
$$

where $z^+ = \sum_{ij} z_{ij}$ , $i = u, v$ and $j = u, v$ .

This ensures that the estimated Goodman's CIs of HIV transmission flows within and between groups/strata in the trial population reflect that amount of information in the sample.

Let the weighted counts of the probable male-female transmission pairs identified within and between intervention and control communities, $\left(z^+\hat{\theta}_{uu}, z^+\hat{\theta}_{uv}, z^+\hat{\theta}_{vu}, z^+\hat{\theta}_{vv}\right)$ denote the observed cell frequencies, $\delta_1, \delta_2, \cdots, \delta_k$ of a sample of size, $z^+$ from a multinomial distribution with population parameters, $\pi_1, \pi_2, \cdots, \pi_k$, and $l = 1, 2, \cdots, k$ population strata, respectively.

For example, $\delta_l$ represents the weighted counts of probable male-female transmission pairs identified in the $l^{th}$ stratum, and $\pi_l$ the corresponding probability that a probable male-female transmission pair falls within the $l^{th}$ stratum, respectively. In this case, we have $k = 4$ population strata that represent HIV transmission pairs within intervention communities; within control communities; transmission pairs for whom a member of an intervention community was the inferred source of transmission to a control community member and vice-versa. The point estimate for, $\pi_l$ or estimated proportion of HIV transmissions in the trial population in the $l^{th}$ stratum is $\hat{\pi}_l = \frac{\delta_l}{z^+}$. The set of $k$ simultaneous CIs at the 5% significance level for the $k$ population-level HIV transmission flows, $\pi_1, \pi_2, \cdots, \pi_k$ are given by **Cherry, 1996**,

$$\hat{\pi}_l^- \leq \pi_l \leq \hat{\pi}_l^+ \text{ for } (l = 1, 2, \cdots, k) ,$$

where the lower confidence bounds are described by,

$$\hat{\pi}_l^- = \left[B + 2\left(\delta_l - 0.5\right) - \sqrt{B\left(B + 4\left(\delta_l - 0.5\right)\left(z^+ - \delta_l + 0.5\right)/z^+\right)}\right] / 2\left(z^+ + B\right)$$

and the upper confidence bounds are described by,

$$\hat{\pi}_l^+ = \left[B + 2\left(\delta_l + 0.5\right) + \sqrt{B\left(B + 4\left(\delta_l + 0.5\right)\left(z^+ - \delta_l - 0.5\right)/z^+\right)}\right] / 2\left(z^+ + B\right) .$$

Here, $B$ denotes the 95th percentile of a chi-squared distribution with $k - 1$ degrees of freedom. Note that $\hat{\pi}_l^- = 0$ if $\delta_l = 0$, and $\hat{\pi}_l^+ = 1$ if $\delta_l = z^+$.

We have implemented the algorithms to estimate HIV transmission flows within and between population groups accounting for sampling heterogeneity; and the corresponding CIs as an R package, *bumblebee* that will be made available at the following URL: https://magosil86.github.io/bumblebee/. A step-by-step tutorial on how to estimate HIV transmission flows with *bumblebee* and accompanying example data sets can be accessed at the following URL: https://github.com/magosil86/bumblebee/blob/master/vignettes/bumblebee-estimate-transmission-flows-and-ci-tutotial.md (**Magosil, 2021**).

## Calculation of the weighted mean age gap between males and females in inferred probable transmission pairs accounting for variability in sampling of the trial population by gender and 5-year age group

### Overview

To compare the ages of males and females in inferred male-female probable transmission pairs we computed weighted (arithmetic) mean age-gaps between males and females in (1) all inferred probable transmission pairs (n = 82), (2) male-to-female transmission events (n = 45), and (3) female-to-male transmission events (n = 37).

### Data set and sampling probabilities

Consider a deep-sequence phylogenetic analysis where $L_{mf}$ male-female probable transmission pairs/events were phylogenetically inferred in the sequenced sample. Here, the sequenced sample refers to individuals whose HIV-1 viral genomes were successfully sequenced and met minimum criteria for inclusion in phylogenetic analysis. Let individuals in the sequenced sample be grouped by gender into 5-year age-categories according to their age at enrollment in the study so that the number of successfully sequenced males in a specific 5-year age category is denoted by, $n_{m(\text{5-year age group})}$ and the number of successfully sequenced females, $n_{f(\text{5-year age group})}$. Similarly, let $N_{m(\text{5-year age group})}$ and $N_{f(\text{5-year age group})}$ denote the estimated number of males and females respectively with HIV in the trial population in a specific 5-year age category. So that the sampling probabilities of male and female partners in a

probable transmission pair are given by, $s_{m(\text{5-year age group})} = n_{m(\text{5-year age group})} / N_{m(\text{5-year age group})}$ for the male partner and, $s_{f(\text{5-year age group})} = n_{f(\text{5-year age group})} / N_{f(\text{5-year age group})}$ for the female partner.

## Computation of mean weighted age gap

Let $g_i$ denotes the age gap in years between male and female partners in the $i^{th}$ transmission pair (i.e., male age – female age) and $w_i$ represents the weight assigned to the $i^{th}$ transmission pair. The weight of a male-female probable transmission pair is computed as the inverse of the product of the sampling probabilities of the male and female partners in the pair,

$$w_i = \frac{1}{s_{m(\text{5-year age group})} * s_{f(\text{5-year age group})}}. \tag{14}$$

Then the weighted mean age gap is given by

$$\bar{g} = \frac{\sum_{i=1}^{n} w_i * g_i}{\sum_{i=1}^{n} w_i}, \tag{15}$$

which can be expanded as:

$$\bar{g} = \frac{w_1 g_1 + w_2 g_2 + \cdots + w_n g_n}{w_1 + w_2 + \cdots + w_n}. \tag{16}$$

Estimation of the weighted age gap variance: we compute the variance of the age-gaps between males and females in probable transmission pairs as:

$$Var\left(\text{weighted age gap}\right) = \frac{\sum_{i=1}^{n} w_i \left(g_i - \bar{g}\right)^2}{V_1 - \left(\frac{V_2}{V_1}\right)}, \tag{17}$$

where $V_1 = \sum_{i=1}^{n} w_i$ represents the sum of the weights, and $V_2 = \sum_{i=1}^{n} w_i^2$ the sum of the squared weights, respectively.

## Permutation test to evaluate whether there was preferential sexual mixing among trial communities by geographic proximity

To evaluate whether people were more likely to form out-of-community sexual partnerships with partners from nearby communities than communities further away, we performed a permutation test under the null hypothesis that the mean travel distance (kilometers) and drive time (hours) between communities where HIV transmission events were identified was no different from that between any pair of randomly selected trial communities. We obtained null distributions for the mean travel distance and drive time, respectively by permuting the order of community pairs over 10,000 iterations. The underlying correlation structure of viral genetic linkage was preserved, for example, transmission events that occurred between members of two communities or between members of a community and two or more other communities. A one-sided *p* value was obtained as the fraction of iterations where the mean travel distance between permuted community pairs was smaller than that for community pairs where HIV transmission events were identified; similarly, a one-sided *p* value was derived for travel time. Travel distances and drive times were sourced from the google distance matrix API (application programming interface) with the mapsapi R package v0.4.2. and will be made available as an R data set. The permutation-test was performed using the R statistical software version 3.5.2.

## Results

### Consensus sequence phylogenetics to identify clusters of participants with genetically similar HIV-1 infections

#### Most participants with HIV at time of enrollment in the trial were on antiretroviral treatment and virally suppressed

Out of the 5114 participants who provided a sample for HIV viral genotyping, 3178 were sampled at baseline and 1936 were sampled post-baseline, that is, about a year or more after the end of baseline survey activities in their communities. At the time of identification, most (86%, *n* = 4410) individuals with HIV-1 infection were on antiretroviral therapy and virally suppressed (≤400 copies per ml of blood plasma). Compared to the 2011 Botswana census population of trial communities, our data set underrepresents men (n = 1475 vs. 3639) and individuals aged 16–24 years (*Supplementary file 1-Table 1A*).

#### Most of the HIV viral genomes sampled were from prevalent infections, and typical genetic distances were large

Out of the 5114 participants whose HIV-1 viral genomes were deep-sequenced, 3832 participants met minimum criteria for inclusion in phylogenetic analyses (see Materials and methods); and of those

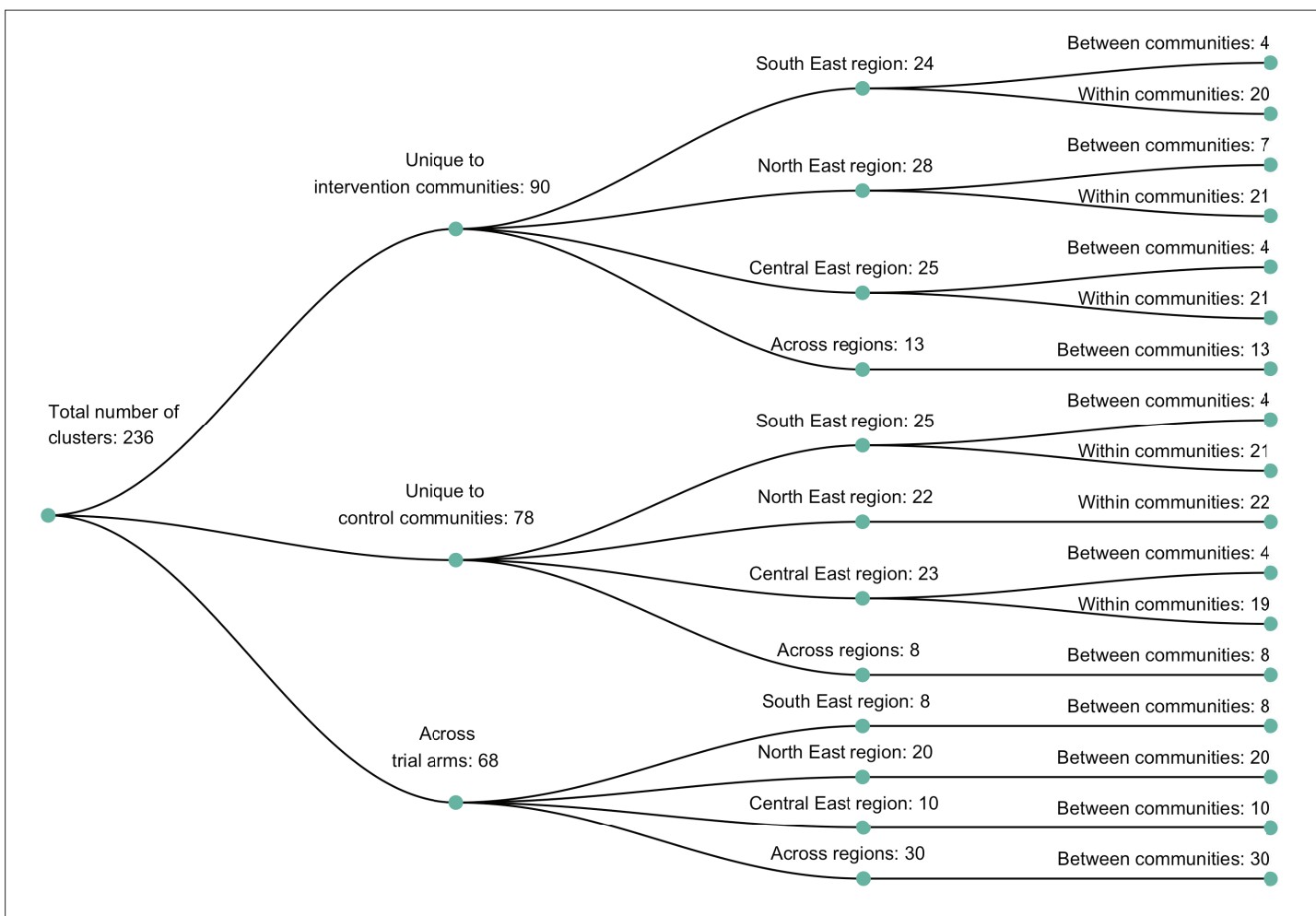

**Figure 5.** A dendogram showing clusters of genetically similar HIV-1 infections in the Botswana/Ya Tsie trial. Genetic clusters are summarized by randomized-HIV-intervention condition, geographical region, and occurrence within and between trial communities. Clusters were identified from 3832 HIV-1 viral whole-genome consensus sequences as two or more sequences separated by a genetic distance not exceeding 0.045 (4.5%) substitutions per site and a bootstrap support threshold of at least 80%. More genetic clusters occurred within trial communities and geographical regions than between them.

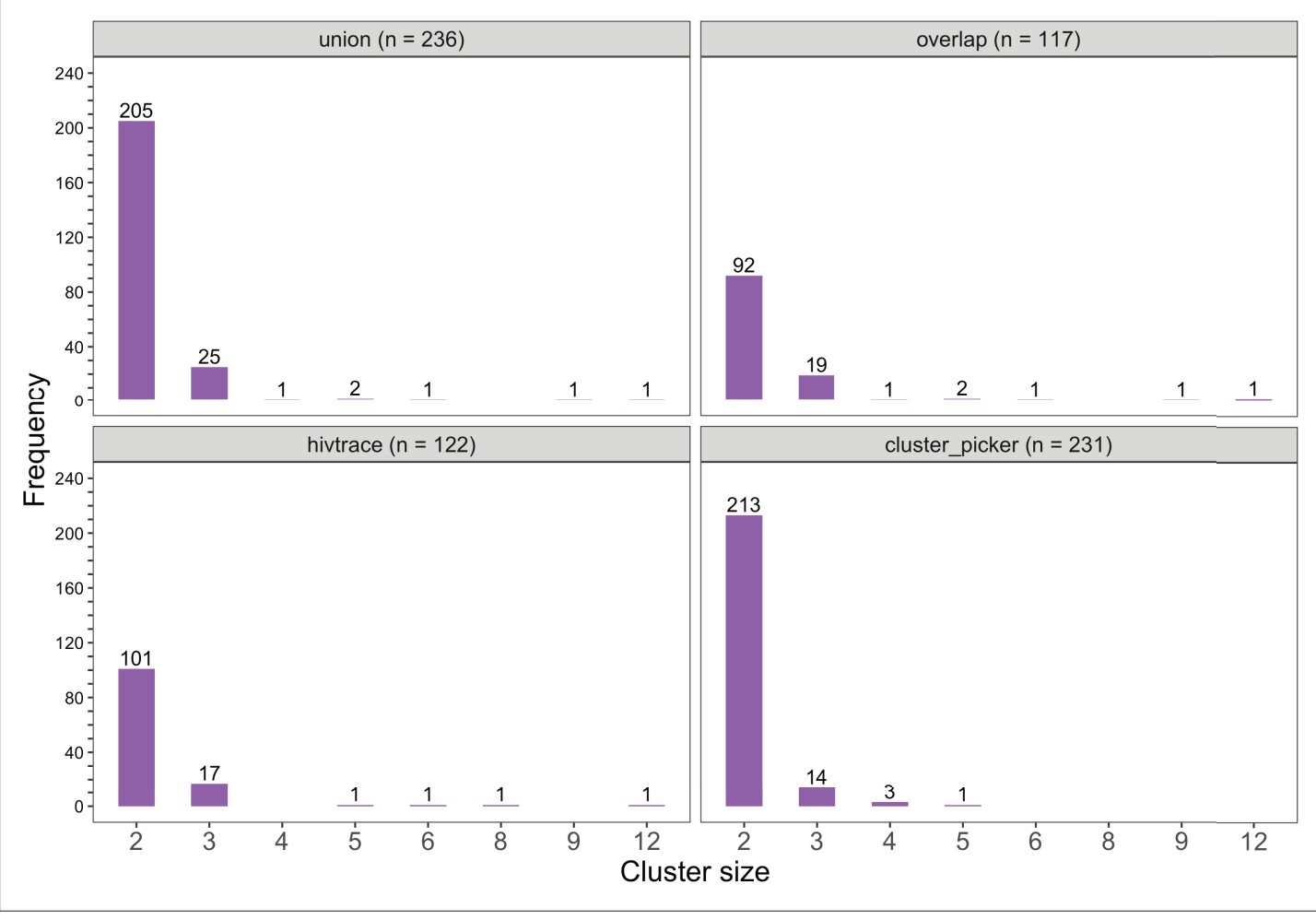

**Figure 6.** Barplots showing the size distribution of clusters of genetically similar HIV-1 infections in the Botswana/Ya Tsie trial. The bottom two plots show the size distribution of genetic clusters identified with HIV-TRACE (n = 122) and Cluster Picker algorithms (n = 231), respectively; and the top two plots show the size distribution of the union (n = 236) and overlap (n = 117) of genetic clusters detected with HIV-TRACE and Cluster Picker. Most clusters were small-sized comprising two or three members.

3832 participants, 2465 were sampled at baseline and 1367 sampled post-baseline. The mean ± standard deviation pairwise genetic distance between HIV-1 viral whole-genome consensus sequences of participants included in phylogenetic analyses was 12.5 ± 3.0%, and the maximum pairwise distance was 40% substitutions per site (*Figure 3A*) (see Materials and methods on comparing genetic distances between HIV-1 viral consensus sequences of trial participants included in phylogenetic analyses). The large distances between sequences possibly reflect sparse sampling and/or the sampling of a large percentage of trial participants relatively late in infection. HIV-1 viral populations typically diverge over time owing in part to within-host evolution.

### About one in seven genotyped participants included in consensus sequence phylogenetic analyses were assigned to genetic similarity clusters of people with closely related HIV-1 infections

We identified 236 genetic similarity clusters comprising HIV-1 viral consensus sequences from 14% (525 / 3832) of the participants (*Figure 5*) (see Materials and methods section on consensus sequence phylogenetics to identify clusters of participants with genetically similar HIV-1 infections). This represents the union of clusters detected with HIV-TRACE (122 clusters, comprising 283 consensus sequences) and Cluster Picker (231 clusters, comprising 484 consensus sequences) algorithms (*Figure 6*). To maximize the number of transmission pairs that could be identified between participants in the Botswana/

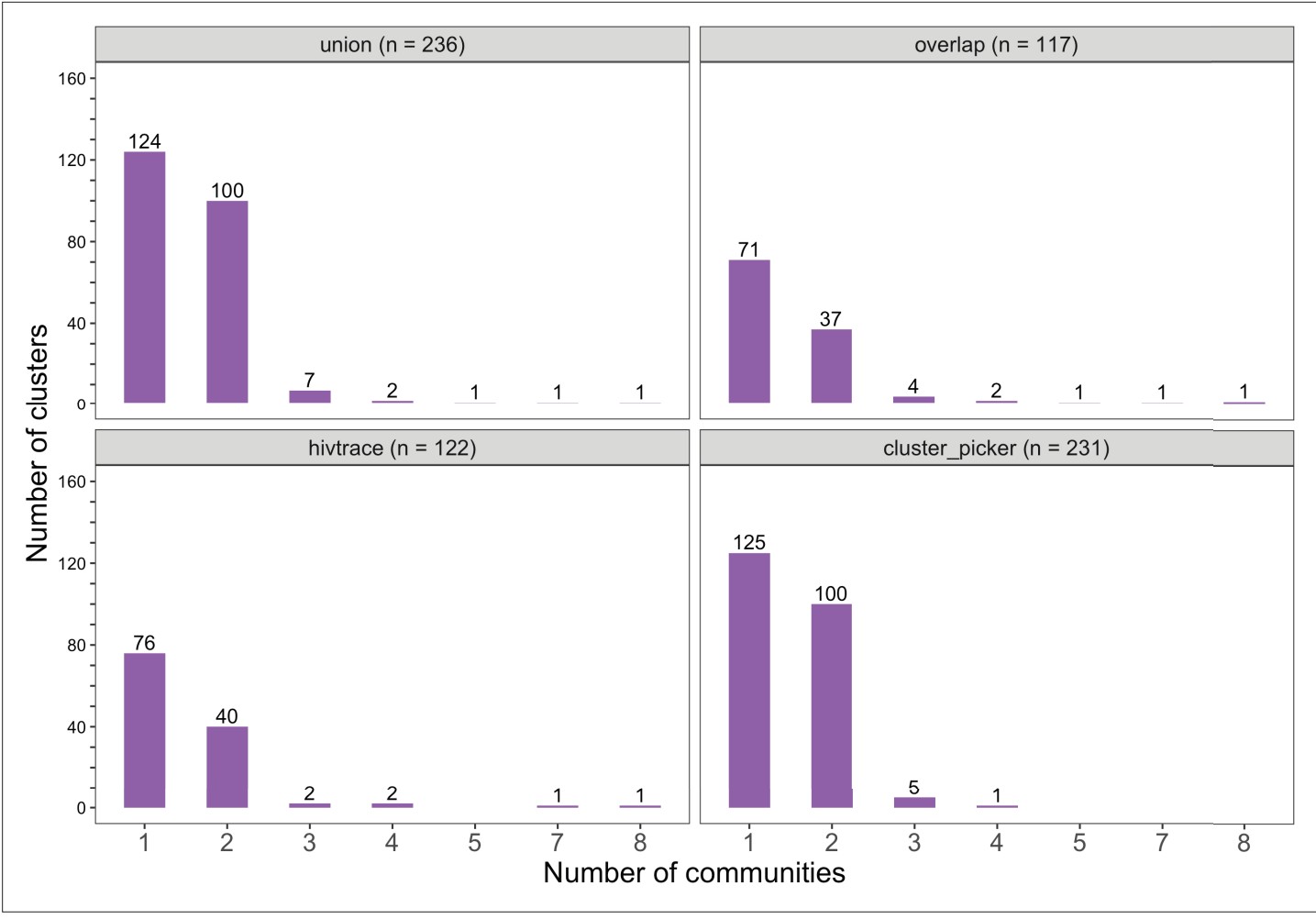

**Figure 7.** Barplots showing the spread of genetic clusters across trial communities. The bottom two plots show the number of genetic clusters identified with HIV-TRACE (*n* = 122) and Cluster Picker algorithms (*n* = 231), respectively; and the top two plots show the union (*n* = 236) and overlap (*n* = 117) of genetic clusters found with HIV-TRACE and Cluster Picker. Most lineages localized to one or two trial communities; with few clusters having members spread out across five or more trial communities.

Ya Tsie trial with Phyloscanner we used the union of clusters detected with HIV-TRACE and Cluster Picker. There was substantial overlap between the two clustering algorithms, 96% (117 / 122) of the genetic clusters found with HIV-TRACE were also detected with Cluster Picker (*Figure 6*). Only 4% (21 / 525) of the participants assigned to genetic similarity clusters were seroconverters from the HIV-incidence cohort (see Materials and methods section on definition of seroconverters).

## Most genetic clusters had few members and limited geographic spread

The sizes of genetic clusters ranged from 2 to 12 members, with 97% (230 / 236) of clusters comprising two or three members (*Figure 6*). The two largest clusters had 9 and 12 members, respectively, none of whom were seroconverters from the HIV-incidence cohort; their members resided in communities in all three geographic regions. Overall, however, genetic clusters were generally concentrated within geographic regions (Central-East, North/North-East, and South-East) than spread out across them (*Figure 5*), with 42% (100 / 236) of clusters localized to two communities (*Figure 7*) and 52% (124 / 236) of the clusters entirely within a single community (*Figure 7*). Moreover, genetic clusters were relatively evenly spread across trial arms; 38% (90 / 236) were unique to intervention communities; 33% (78 / 236) to control communities; and 29% (68 / 236) had genetic cluster members from both intervention and control communities (*Figure 5*). The number of genetic clusters identified within trial communities generally increased with sampling density; for example, communities such as: Shakawe,

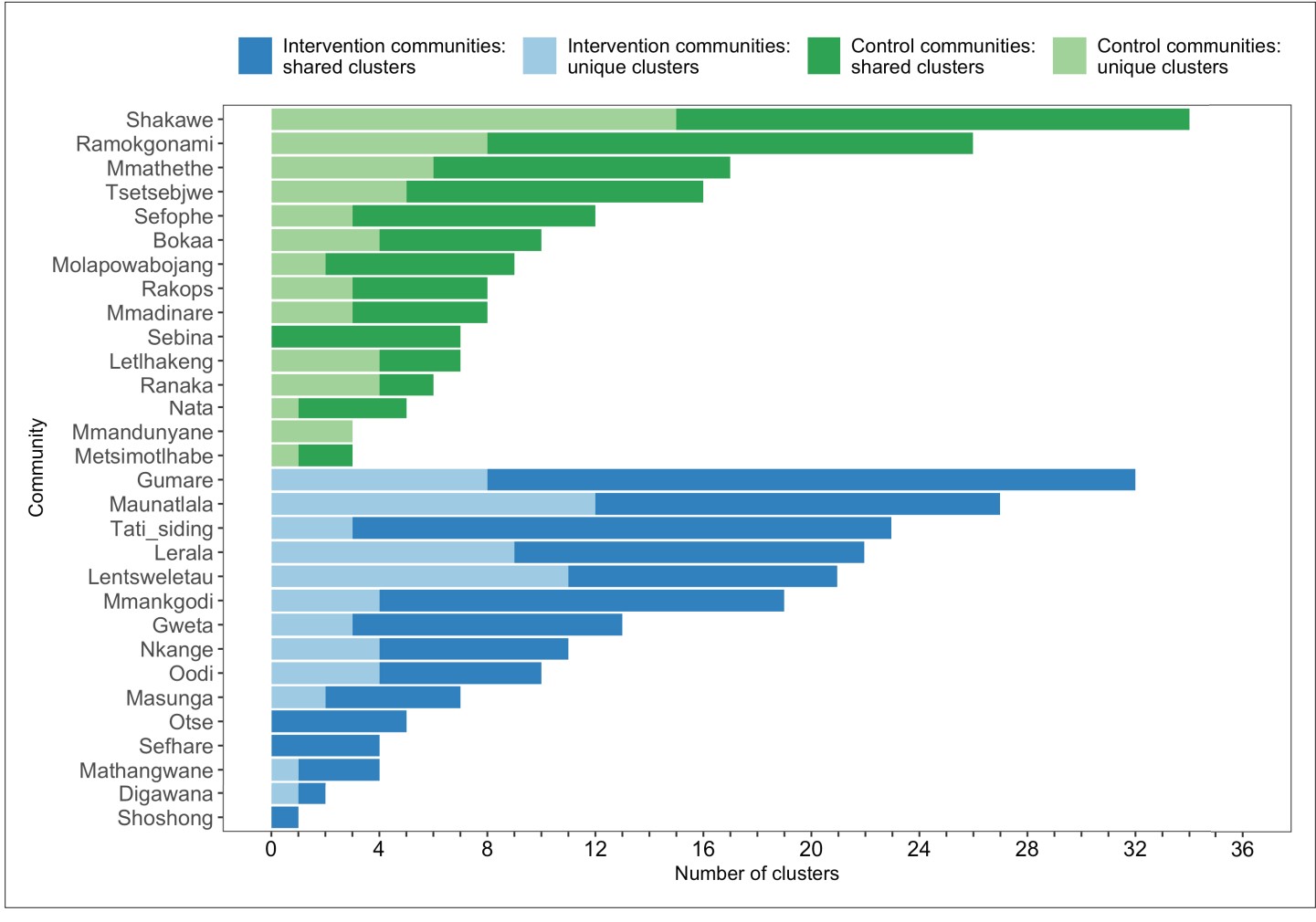

**Figure 8.** A barplot summarizing clusters of genetically similar HIV-1 infections by trial community. Bars represent the total number of genetic clusters identified in each trial community partitioned into clusters that are specific to a community (lighter shade) and those that are shared with at least one other trial community (darker shade). Genetic clusters in intervention communities are shown in blue and those in control communities are represented in green. The number of genetic clusters identified among intervention and control communities varied.

Ramokgonami, Mmathethe, Gumare, Maunatlala, and Tati Siding which had larger numbers of unique and shared clusters compared to other trial communities were also among the most densely sampled (**Figure 8**, **Supplementary file 1-Table 1C**). The abundance of small-sized genetic similarity clusters may reflect a moderate (14%) sampling fraction (**Murray and Alland, 2002**).

## Characteristics of participants in clusters

About 70% (366 / 525) of the HIV-1 viral consensus sequences in clusters were from women (**Figure 9**). This result was consistent with the overrepresentation of women among sequenced infections (**Supplementary file 1**, Table 1A). The ages of clustered men and women were similar with women having a median [lower–upper quartile] age of 37.8 [29.6 – 45.1] years and men 43.0 [34.9 – 51.0] years at time of sampling. 33% (78 / 236) of clusters included at least one participant with unsuppressed virus. Additionally, there were few clusters (4% (10 / 236)) where all members had HIV-1 infections with unsuppressed virus (**Supplementary file 1-Table 1D**). 54% (286 / 525) of the participants in clusters of genetically similar HIV-1 infections were also part of a baseline survey of 20% of households randomly sampled from each trial community to establish an incidence follow-up cohort and gather information on different socio-demographic variables including sexual risk behavior; see **Makhema et al., 2019**; **Gaolathe et al., 2016** for details. Among them, 10% (30 / 286) of participants self-reported having multiple partners in the past twelve months; 73% (208 / 286) of participants self-reported a single

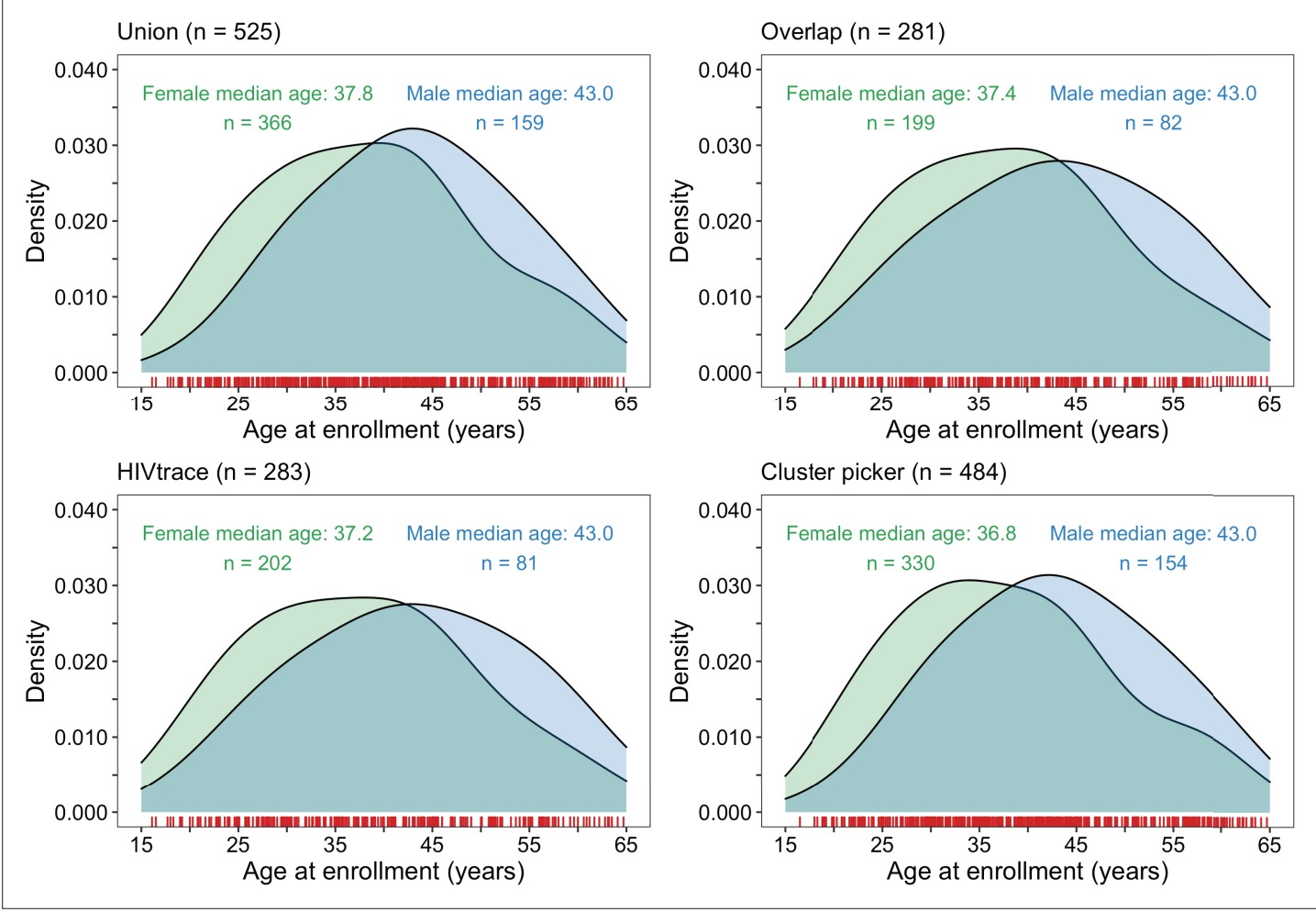

**Figure 9.** Density plots summarizing trial participants in clusters of genetically similar HIV-1 infections by age and gender. The top two panels show the age-gender distribution of trial participants whose HIV-1 viral whole-genome consensus sequences were in the union (n = 525) and overlap (n = 281) of genetic clusters identified with HIV-TRACE and Cluster Picker. The bottom two plots show the age-gender distribution of trial participants whose HIV-1 viral whole-genome consensus sequences were in genetic clusters identified with HIV-TRACE (n = 283) and Cluster Picker (n = 484) clustering algorithms. The raw data is shown as a rug plot (red tick marks) on the x-axis. Men and women in genetic clusters had similar ages.

partner in that period, 15% (44 / 286) of participants self-reported zero partners in that period and the rest did not disclose their number of partners. Of the 30 participants involved in multiple relationships, 90% (27 / 30) disclosed having one other sexual partner, and 10% (3 / 30) reported having two other sexual partners in addition to their primary sexual partner.

## Deep-sequence phylogenetics to infer the probable order of transmission events within identified clusters of genetically similar HIV-1 infections

We identified 153 highly supported probable source-recipient pairs within 236 clusters of genetically similar HIV-1 infections. Seventy-one were same-sex linkages between women (n = 65) or men (n = 6), and the remaining were male-female pairs (n = 82). Considering that the predominant mode of HIV-1 transmission in Botswana and most of southern Africa is through heterosexual contact, that direct transmission between women is rare (*Chan et al., 2014*), same-sex pairs were presumed to be members of transmission chains with unsampled intermediates. We used the probability of inferring a phylogenetically linked directed male-male pair in the sample to calibrate an upper bound on the FDR of inferring direct transmission between males and females in phylogenetically linked male-female pairs; the estimated number of linked male-female pairs in the sample with unsampled intermediates

was approximately 30, corresponding to a FDR of 36% (30 / 82) (see Materials and methods section on estimating error rates in phylogenetic inference of direct transmission between sampled males and females). This estimated false positive rate is likely inflated given that two individuals would need to be missing from the sequenced sample to incorrectly infer transmission in a male-female pair, whereas only a single female would need to be missing to erroneously infer transmission in a male-male pair. As indicated earlier (see Materials and methods section on criteria for inclusion in phylogenetic analyses), we expected our data set to contain the HIV-1 viral sequence of the transmitter for 11% (0.14 × (1 − 0.21)) of sequenced participants included in phylogenetic analyses (n = 3832), compared with the 82 male-female probable transmission pairs that we identified thus suggesting that consensus sequence phylogenetic analysis missed some viral genetic linkages between participants with chronic infections; and deep-sequence phylogenetic analysis identified only a subset of HIV-1 transmission events between participants assigned to genetic clusters. We restricted further analyses on inference of HIV transmission patterns to the highly supported 82 male-female probable transmission pairs. Analyses are presented first without an adjustment for sampling variability to illuminate patterns of viral transmission within the deep-sequenced sample. Thereafter an adjustment for variable sampling rates by demographic group (age, gender, trial community, and geographical region) or randomized-HIV-intervention (trial arm) is made to estimate the flow of HIV transmissions within the trial population (see Materials and methods sections on computing age-gender estimates of the number of people with HIV in each trial community and adjustment for variable sampling rates across different demographic groups or randomized-HIV-interventions).

## Age and sex distribution of sources and recipients in inferred transmission pairs

The age distribution of transmitters and recipients, by gender, for the 82 male-female probable transmission pairs is shown in *Figure 10*. Inferred male-female and female-male HIV transmissions had similar age distributions for the inferred source of infection. Males were a mean ± standard deviation of 3.5 ± 9.6 years older than females in probable transmission pairs overall, with a difference of 4.9 ± 7.5 in male-to-female transmission events, and 1.8 ± 11.6 years in female-to-male transmission events. Adjusted for variability in sampling of the trial population by gender and 5-year age group, these differences were somewhat reduced: males in probable transmission pairs were a mean ± standard deviation of 1.3 ± 10.4 years older than females overall; with a difference of 3.8 ± 6.9 years in male-to-female transmission events, and −1.3 ± 12.7 years in female-to-male transmission events (see Materials and methods section on calculation of the weighted mean age gap between males and females in inferred probable transmission pairs accounting for variability in sampling of the trial population by gender and 5-year age group). The difference between the unadjusted and the weighted mean age gap between males and females in probable transmission pairs is consistent with the undersampling of younger males. Quantiles of age differences between males and females in inferred transmission pairs before and after adjustment for variability in sampling are presented in *Supplementary file 1-Table 1E*.

## Most HIV transmissions occurred between similarly aged partners

*Supplementary file 1-Table 1F* shows the proportions (unadjusted) of viral transmission events within and between 5-year age groups among the 82 probable male-female pairs inferred from the deep-sequenced sample. After adjusting for differential sampling by 5-year age group, the predicted proportions of HIV-1 transmissions in the trial population suggested preferential sexual mixing between similarly aged partners (*Figure 11* and *Supplementary file 1-Table 1F*) (see Materials and methods sections on computing age-gender estimates of the number of people with HIV in each trial community and adjustment for variable sampling rates across different demographic groups or randomized-HIV-interventions). An adjustment by both 5-year age group and gender revealed broadly similar transmission patterns in the trial population (*Figure 12* and *Supplementary file 1-Table 1G*).

## Similar proportions of men and women as inferred sources of transmission

55% [95% CI: 39.8 – 69.1] of the 82 inferred male-female transmission events originated from men with the remainder occurring from women, 45% [30.9 – 60.2] (*Figure 13*). An adjustment for variable

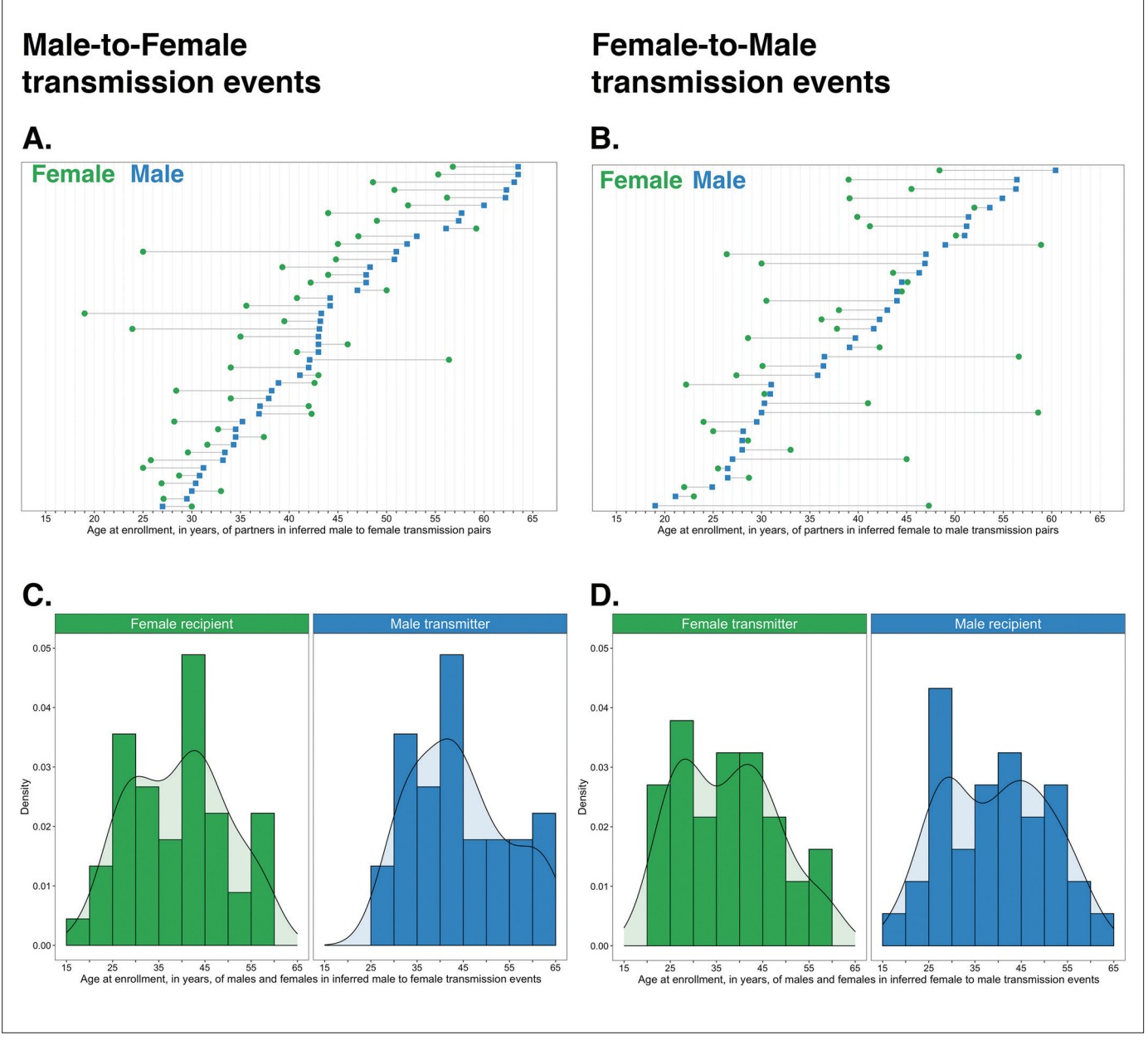

**Figure 10.** Age distribution of 82 male-female probable transmission pairs sampled in the Botswana/Ya Tsie trial. Forty-five male-to-female and 37 female-to-male transmission events were identified from the deep-sequenced viral whole genomes of 525 trial participants in clusters of genetically similar HIV-1 infections. Men were typically older than women in both male-to-female and female-to-male transmission events.

sampling by gender revealed similar transmission patterns in the trial population (***Supplementary file 1-Table 1H***).

## Most inferred HIV transmissions occurred within communities or between neighboring communities

Trial communities could be broadly grouped into three geographical areas: in the North/North-East, South-East, and Central-East region of Botswana, which borders South Africa and Zimbabwe (***Figure 1***). The proportion of inferred transmission events of HIV-1 infection in the deep-sequenced sample between members of the same trial community was 69% (57 / 82) compared with 21% (17 / 82) for members of different trial communities in the same region, and 10% (8 / 82) for members of trial communities in different regions (***Figures 13 and 14***). Of the 74 (74 / 82) transmission events within

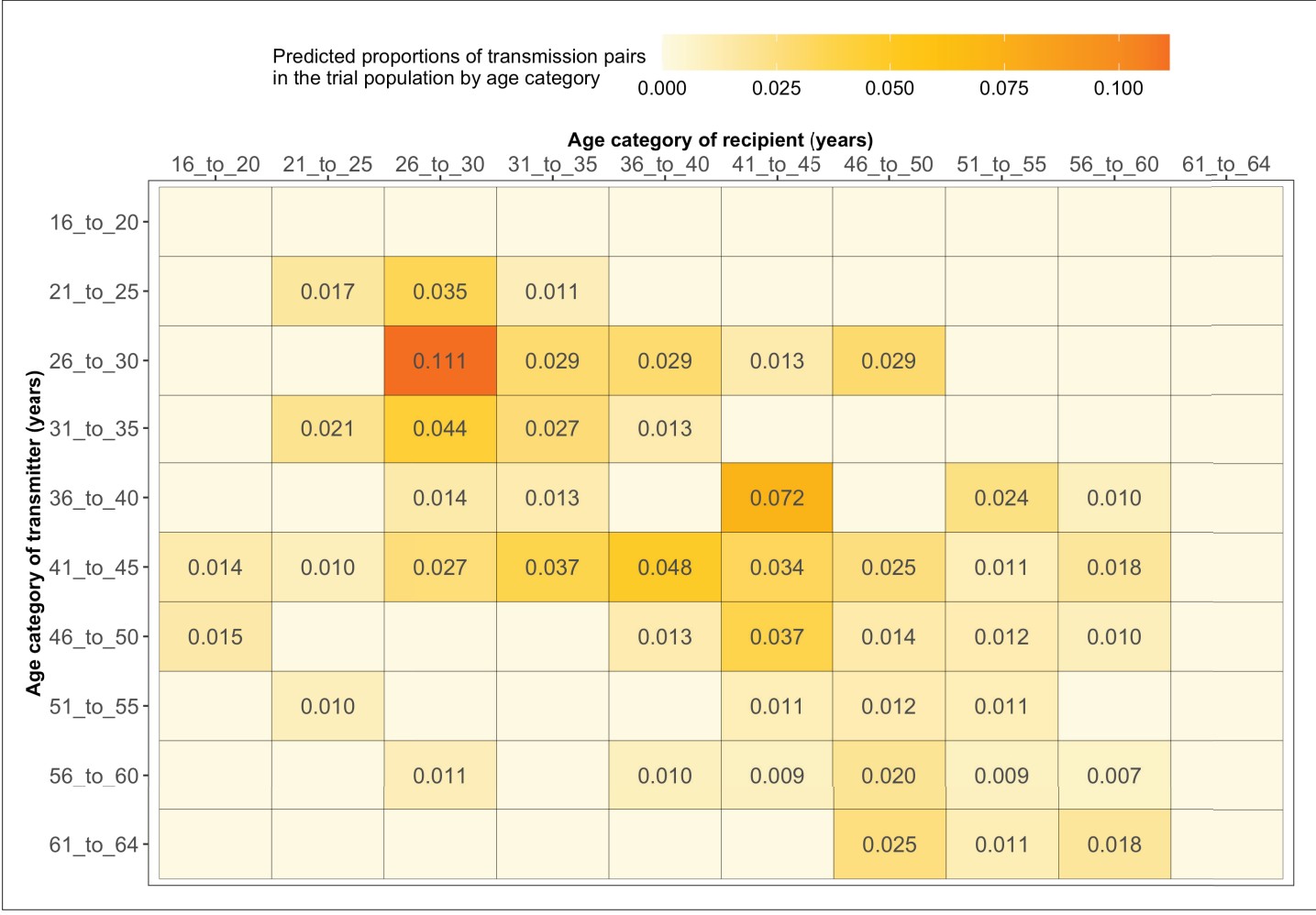

**Figure 11.** Estimated transmission flows of HIV-1 infection within and between 5-year age groups in the Botswana/Ya Tsie trial population. Transmission flows of HIV-1 infection in the trial population were estimated from 82 male-female probable transmission pairs identified from the deep-sequenced HIV-1 viral whole genomes of trial participants ($n$ = 525) in clusters ($n$ = 236) of genetically similar HIV-1 infections. Furthermore, transmission flows were adjusted for differential sampling among age groups (see Materials and methods section on adjustment for variable sampling rates across different demographic groups or randomized-HIV-interventions). Most viral transmission events occurred between similarly aged partners.

a single region (including those within a single community), 22 were in the Central-East, 28 in the North/North-East, and 24 in the South-East (*Figure 13*). For pairs where transmission events linked people in different trial communities the median [lower–upper quartile] driving distance between the trial communities was 161 km [108 – 420 km] or 100 mi [67 – 261 mi] and the median drive time was 1.86 hr [1.41 – 4.68 hr]. Furthermore, a comparison of mean drive times between pairs of trial communities for which transmission events were identified ($n$ = 22) with those of all other possible pairs of participating trial communities excluding same community pairs ($n$ = 900 – 22 – 30 = 848) revealed a shorter drive time on average between genetically linked trial communities compared with what would be expected under random sexual mixing (permutation-test, $p < 0.01$) (see Materials and methods section on permutation test to evaluate whether there was preferential sexual mixing among trial communities by geographic proximity). These results suggest that out-of-community sexual partnerships were more likely to form between partners residing in nearby communities compared with more distant communities.

After adjusting for differential sampling by community, an estimated 24% of transmissions to a resident of a trial community originated from another trial community (*Supplementary file 1-Table 1I*) (see Materials and methods section on adjustment for variable sampling rates across different demographic groups or randomized-HIV-interventions). This is consistent with the high levels of mobility in Botswana, where people typically have family ties in ancestral villages and work in peri-urban or urban

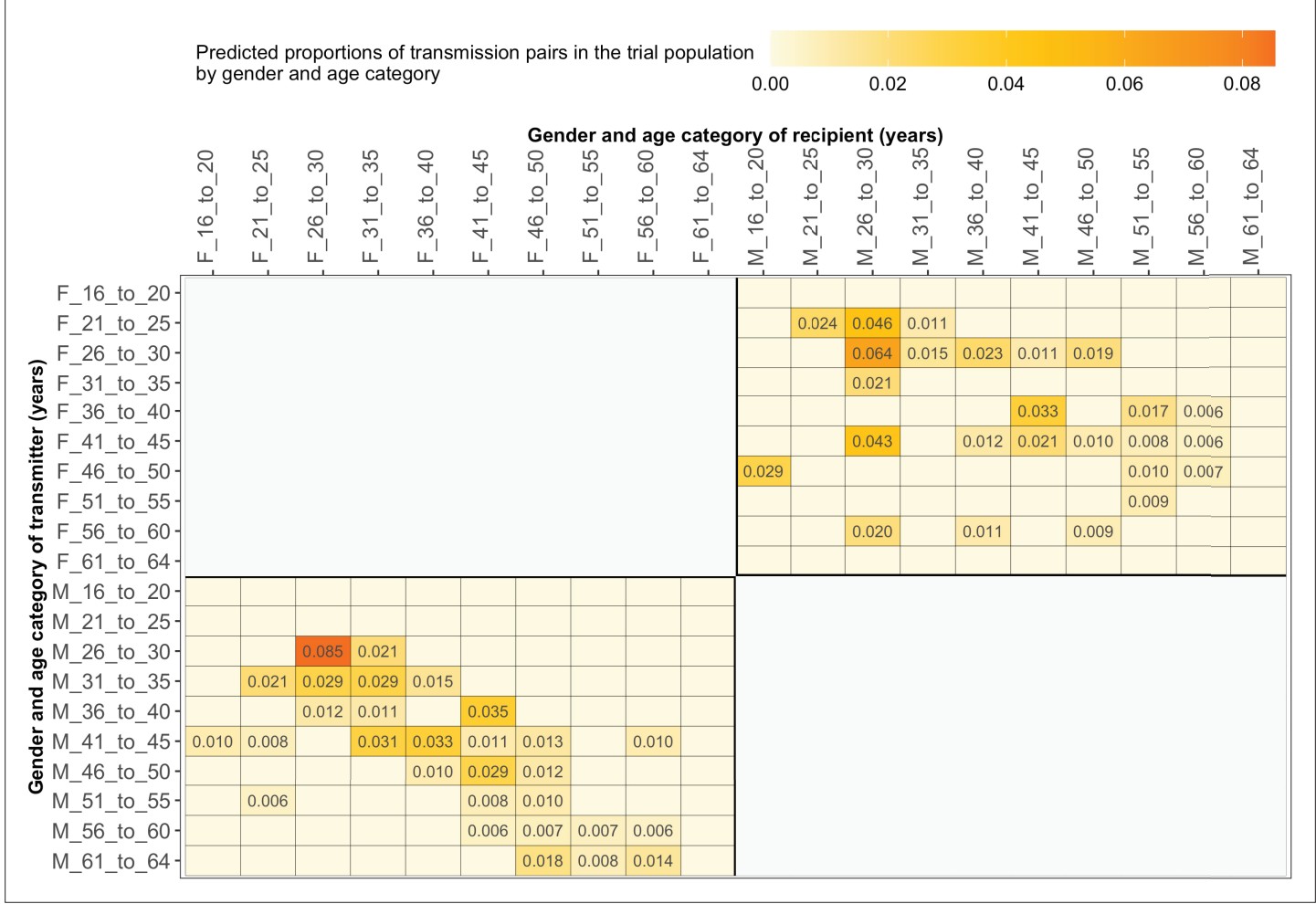

**Figure 12.** Estimated transmission flows of HIV-1 infection by gender within and between 5-year age groups in the Botswana/Ya Tsie trial population. Transmission flows were estimated as described in *Figure 11*. Additionally, transmission flows were adjusted for differential sampling by gender and age group.

areas (*Essex et al., 2019*). Furthermore, trial communities were distributed along two principal highways that might have contributed to increased mobility: the A1 highway that connects the Northern and Southern parts of the country and the A3 highway that branches from the A1 highway connecting the North-Eastern and North-Western regions.

The estimated proportion of HIV transmissions in the trial population, after adjusting for differential sampling by geographic region, that occurred within the South-East region, 50% [95% CI: 29.6 – 70.5] was higher compared to that in the Central-East, 23% [9.7 – 45.4] and North-East regions, 18% [6.5 – 39.8], respectively (*Figure 15A* and *Supplementary file 1-Table 1J*) (see Materials and methods section on adjustment for variable sampling rates across different demographic groups or randomized-HIV-interventions). For comparison, the proportion of trial participants with viral genomes that met criteria for inclusion in phylogenetic analyses were 26% (1000 / 3832) in the South-East region, 42% (1630 / 3832) in the North/North-East region, and 31% (1202 / 3832) in the Central-East region (*Supplementary file 1-Table 1C*). The South-East region includes Gaborone city, the economic and administrative capital of Botswana; conversely, economic activity in the North-East and Central-East regions is centered around mining and agriculture.

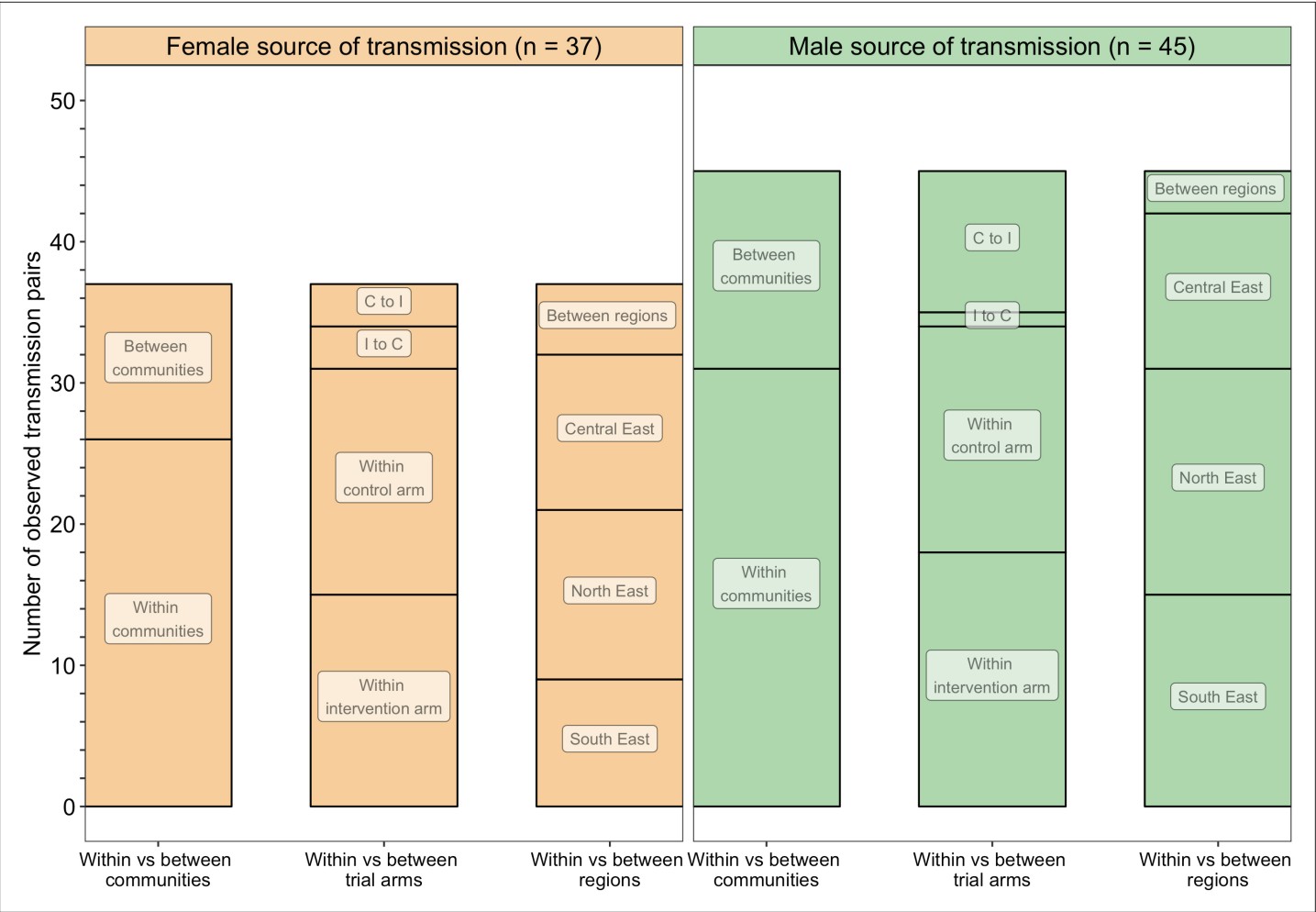

**Figure 13.** A barplot showing counts of male-female probable transmission pairs sampled in the Botswana/Ya Tsie trial by geographical location. The 82 probable male-female infector-infectee pairs that were identified from clusters of genetically similar HIV-1 infections (*n* = 236) in the Botswana/Ya Tsie trial are first summarized into those with a male versus a female source of viral transmission. Male-to-female and female-to-male transmission events are then further partitioned according to whether partners in a transmission pair resided in the same or different trial communities and if those trial communities belonged to the same or different trial arms and geographical regions. C to I denote transmission into an intervention community from a control community and I to C represent transmission into a control community from an intervention community. Male-to-female transmission events are shown in green and female-to-male transmission events are represented in yellow. About 69% (*n* = 57) of the sampled transmission events in the Botswana/Ya Tsie trial occurred within the same trial community.

## More HIV transmission events to residents of intervention communities originated from residents of control communities than vice versa

### Overall analysis

The proportion of inferred transmission events of HIV-1 infection in the deep-sequenced sample occurring within intervention (*n* = 33) or within control communities (*n* = 32) was 79% (65 / 82) compared with 21% (17 / 82) across intervention and control communities (*Figures 13 and 14*). Of the 33 transmission events inferred within intervention communities, 7 occurred between partners residing in different communities; and for those inferred within control communities, 1 of the 32 transmission events occurred between partners from different communities. There were more transmission events identified in intervention communities that originated from control communities than the reverse (*n* = 13 vs. 4).

After adjusting for differential sampling by trial arm, the estimated proportions of HIV transmissions in the trial population within control communities, 51% [35.9 – 66.5] was about twice that predicted in intervention communities, 29% [16.6 – 44.6] (*Figure 15B* and *Supplementary file 1-Table 1K*); and

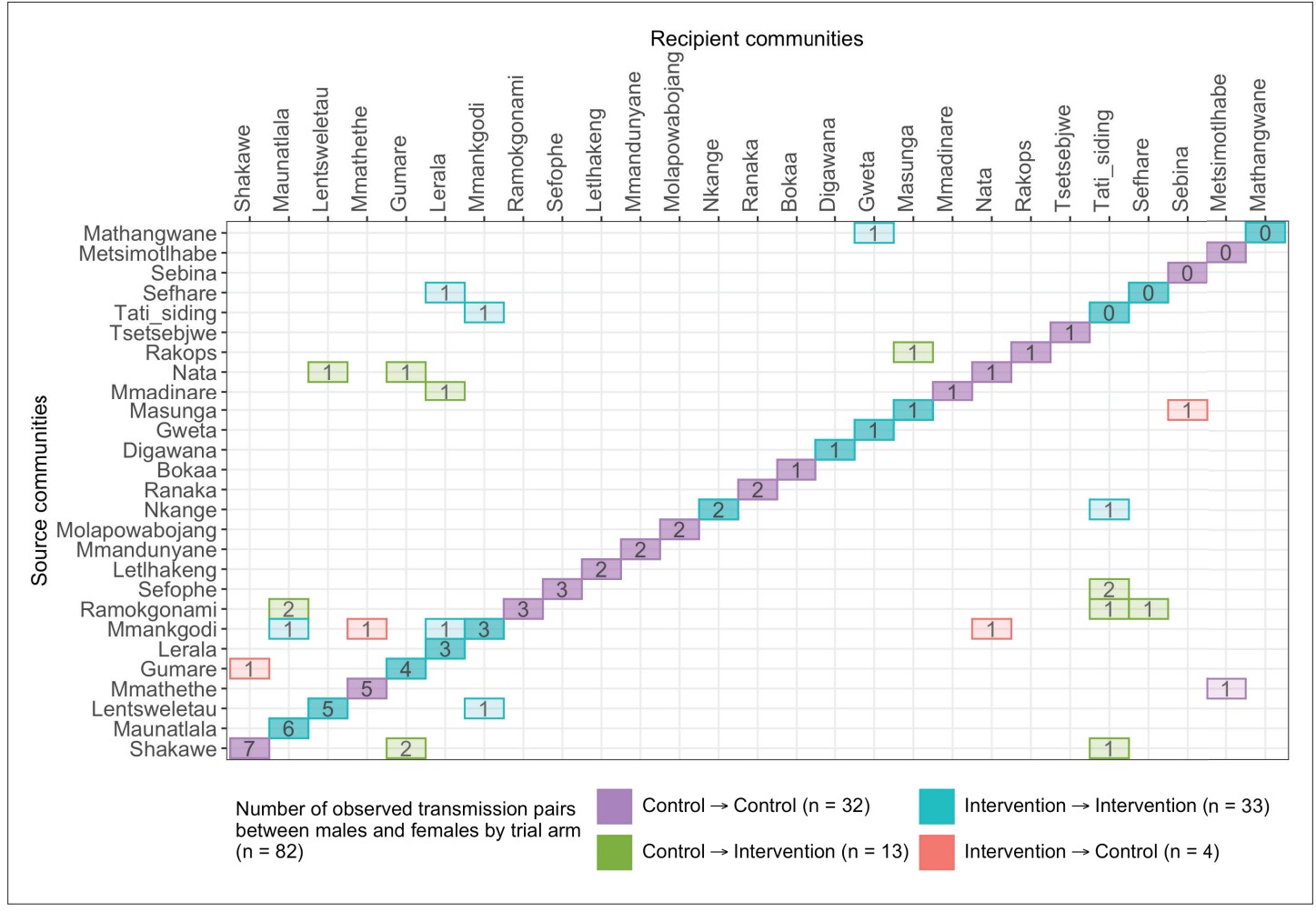

**Figure 14.** A transmission matrix summarizing sampled transmission events within and between Botswana/Ya Tsie trial communities. Numbered tiles represent counts of transmission events identified between a pair of trial communities. There were 32 transmission events identified within control communities (magenta), 33 within intervention communities (cyan), 13 from control-to-intervention communities (green), and 4 from intervention-to-control communities (red). Most transmission events were identified within trial communities. The number of sampled transmission events into intervention communities from control communities was higher than the reverse (n=13 vs. 4).

the estimated proportion of HIV transmissions in the trial population flowing into intervention communities from control communities, 15% [6.9 – 30.1] was about three times higher than the reverse, 5% [1.0 – 16.9] (see Materials and methods section on adjustment for variable sampling rates across different demographic groups or randomized-HIV-interventions). Comparatively, the proportions of trial participants that met criteria for inclusion in phylogenetic analyses from control communities were 40% (1551 / 3832) and from intervention communities, 60% (2281 / 3832).

## Baseline analysis

Of the 82 male-female probable transmission pairs identified in the deep-sequenced sample, 51 involved a recipient that was first sampled during the period of baseline household survey activities in their community, that is, a transmission that could not have been affected by the intervention because it occurred before the start of the intervention. Among these (n = 51), the proportion of inferred transmission events that occurred within intervention (n = 19) or within control communities (n = 25) was 86% (44 / 51) compared with 14% (7 / 51) across intervention and control communities (*Supplementary file 1-Table 1L*). Of the 19 transmission events identified within intervention communities, 4 were between partners living in different communities; compared with 1 out of 25 transmission events

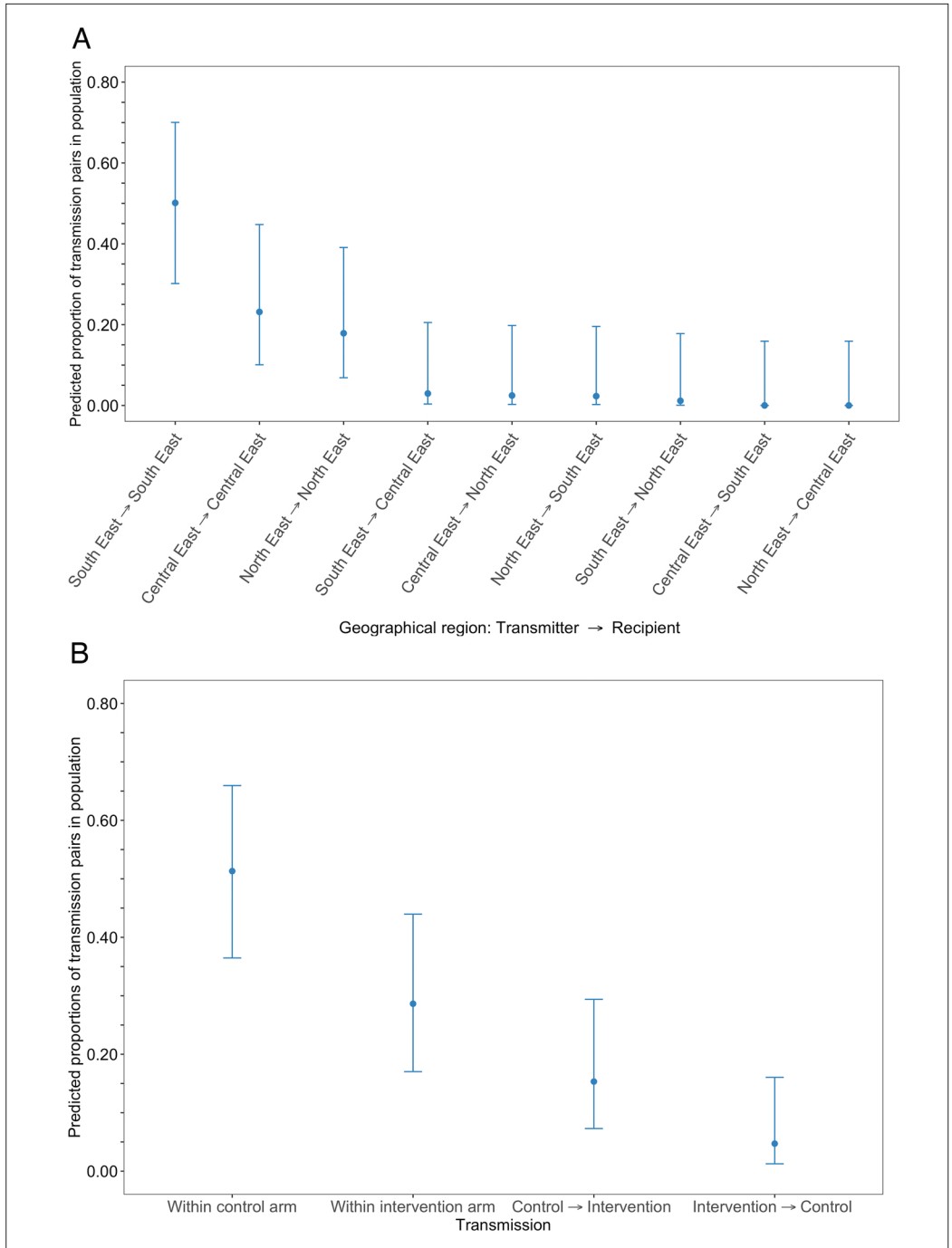

**Figure 15.** Estimated transmission flows of HIV-1 infection in the Botswana/Ya Tsie trial population within and between geographical regions and trial arms. Transmission flows were estimated as described in *Figure 11*. Additionally, transmission flows were adjusted for differential sampling among geographical regions and trial arms respectively. (**A**) Estimated transmission flows of HIV-1 infection in the Botswana/Ya Tsie trial population within and between geographical regions. Communities in the Botswana/Ya Tsie trial were broadly grouped into three geographical regions in the Central-East, North/North-East, and South-Eastern parts of the country. The flow of HIV-1 transmissions in the trial population was highest in the South-East region (50.1%) compared with the Central-East (23.1%) and North/North-East (17.9%) regions, respectively. (**B**) Estimated transmission flows of HIV-1 infection within and between intervention communities and control communities in the Botswana/Ya Tsie trial population. Most transmissions of HIV-1 infection occurred within the same trial arm and the flow of viral transmissions into intervention communities from control communities (15.3%) was about three times higher than the reverse (4.7%).

identified within control communities. The number of transmission events identified in intervention communities that originated from control communities was similar to the reverse (n = 4 vs. 3).

Adjusted for sampling variability among trial arms the flow of HIV transmissions in the trial population within control communities, 62% [41.6 – 78.8] was more than two times that in intervention communities, 25% [11.8 – 45.9] (*Supplementary file 1-Table 1L*); and the flow of HIV transmissions into intervention communities from control communities, 7% [1.5 – 25.3] was similar to the reverse, 5% [0.9 – 22.9] (see Materials and methods section on adjustment for variable sampling rates across different demographic groups or randomized-HIV-interventions).

### Post-baseline analysis

By contrast, 31 of the 82 male-female probable transmission pairs inferred in the deep-sequenced sample involved a recipient that was first sampled about a year or more after the end of baseline household survey activities in their community; this subset reflects transmissions that could have in principle been affected by the intervention, as they may have occurred after the intervention began. Among these (n = 31), the proportion of inferred transmission events that occurred within intervention (n = 14) or within control communities (n = 7) was 68% (21 / 31) compared with 32% (10 / 31) across intervention and control communities (*Supplementary file 1-Table 1M*). Moreover, 3 of the 14 transmission events identified within intervention communities were between partners residing in different communities, and all 7 transmission events identified within control communities were between partners living in the same community. We identified more transmission events in intervention communities that originated from control communities than vice versa (n = 9 vs. 1).

Adjusted for variability in sampling among trial arms the flow of HIV transmissions within control communities, 32% [13.3 – 58.3] was similar to that within intervention communities, 34% [15.0 – 60.9]; and HIV transmissions into intervention communities from control communities, 30% [12.2 – 56.7] were more common than the reverse, 3% [0.1 – 27.3] (*Supplementary file 1-Table 1M*) (see Materials and methods section on adjustment for variable sampling rates across different demographic groups or randomized-HIV-interventions). This result is consistent with a predicted benefit of the treatment-as-prevention intervention in reducing HIV transmission in the Botswana/Ya Tsie trial, albeit precise dates of transmission were not identified.

## A sensitivity analysis to evaluate the impact of probable transmission pairs with unsampled intermediates on the patterns of HIV transmission within and between intervention communities and control communities in the Botswana/Ya Tsie trial

The first set of sensitivity analyses was performed with highly supported directed same- and opposite-sex transmission pairs identified in: (1) HIV-TRACE clusters, (2) Cluster Picker clusters, (3) the overlap of HIV-TRACE and Cluster Picker clusters, and (4) the union of HIV-TRACE and Cluster Picker clusters (*Supplementary file 1-Table 1N*). The second set of sensitivity analyses was performed for the same four categories but restricted to highly supported directed opposite-sex pairs only where the recipient in a transmission pair was first sampled about a year or more after the end of baseline household survey activities in their community (i.e., post-baseline) (*Supplementary file 1-Table 1O*). In both sets of sensitivity analyses, the results were consistent with the primary analysis: transmissions into intervention communities from control communities were more common than the reverse post-baseline (*Supplementary file 1-Table 1N and O*) (see Materials and methods section on adjustment for variable sampling rates across different demographic groups or randomized-HIV-interventions). The signal was stronger as well as being more interpretable in the analysis restricted to opposite-sex pairs compared to the analysis that included same-sex pairs, which likely included one or more unsampled intermediates.

## Discussion

HIV incidence remains high in East and Southern Africa especially among young women. Promising HIV prevention intervention packages that combine population-level HIV testing coupled with strengthened linkage-to-care and early initiation of antiretroviral therapy, jointly termed universal test-and-treat, have yielded lower than expected reductions in the occurrence of new HIV-1 infections

(*Hayes et al., 2019*; *Makhema et al., 2019*; *Havlir et al., 2019*; *Iwuji et al., 2018*; *Abdool Karim, 2019*). This makes it more challenging to attain the UNAIDS goal of ending the HIV/AIDS epidemic as a public health emergency by 2030. An emerging hypothesis among four of the largest universal test-and-treat HIV prevention trials ever conducted in East and Southern Africa in Kenya, Uganda, Botswana, South Africa, and Zambia (*Abdool Karim, 2019*), comprising tens of thousands of participants, is that the intervention impact to reduce population-level HIV incidence was dampened by population mobility; in particular, sexual partnerships formed (1) between individuals in communities randomized to different HIV-intervention conditions and (2) with individuals in communities outside the trial population.

In this study, we focused on one of the four cluster/community-randomized trials, the Ya Tsie trial conducted in Botswana (*Makhema et al., 2019*), to quantify the extent of sexual mixing within and between: trial communities, trial arms, and geographic regions. Furthermore, we quantified the contribution of age-difference and gender to the spread of HIV infections in Eastern Botswana. We first used HIV-1 whole-genome viral consensus sequences to identify clusters of trial participants with genetically similar HIV-1 infections. This was done as a filtering step to save time and computational resources by excluding distantly related sequences. Thereafter, we employed deep-sequence phylogenetics to resolve the probable order of HIV transmission events within each identified cluster. Identified transmission events in the deep-sequenced sample were then used to estimate the flow of HIV transmissions in the trial population adjusting for sampling variability.

We found that HIV transmissions in the trial were more likely to occur within communities or between neighboring communities than between distant trial communities, as well as between similarly aged partners. One of the most striking findings of our analysis was that an estimated 24% of transmissions involving residents of trial communities occurred between different communities in the trial. Given that the trial covered only 7.6% of the national population, this suggests that a large fraction of all transmissions involved a partner outside the trial.

Furthermore, there was substantial sexual mixing between intervention and control communities. Overall, about one in every five HIV transmissions in the trial population involved sexual partnerships between residents of intervention communities and control communities. After baseline, that is, about a year or more after the introduction of the intervention one in three HIV transmissions occurred between intervention and control communities. HIV transmissions into intervention communities from control communities were similar to the reverse at baseline, and ten times more common post-baseline, concordant with a predicted benefit of a universal test-and-treat HIV prevention intervention, though dates of transmission were not identified.

The extent of sexual mixing between residents of intervention and control communities highlights high mobility between the two arms of the Botswana/Ya Tsie trial. For the primary trial endpoint comparing HIV incidence in intervention versus control communities, this kind of mixing will tend to dilute the effect since some of the prevention value of the intervention would appear in control communities, while intervention communities would have incidence that comes from control communities and from communities outside the trial thus is not reachable by the effect of the intervention. Both sources of incidence, from control communities and from outside the trial, could dilute the intervention effect observed in the trial. Although we could not evaluate it in this study, it is possible that out-of-trial index cases would dilute the effect even more strongly than control-arm ones because the control arms comprise only about 5% of Botswana's population, while about 90% of the population lives outside the trial.

These findings might be relevant for the interpretation of the results of the other three universal test-and-treat HIV prevention trials conducted in Kenya, Uganda, South Africa, and Zambia. An important next step would be to quantify the extent to which the size of the observed effect in the Botswana/Ya Tsie trial was reduced by imported infections between partners in communities randomized to different HIV-intervention conditions and partners in communities outside the trial population.

There was a deliberate effort to enroll men and young individuals in the Botswana/Ya Tsie study as these groups are less likely to be diagnosed with HIV and engaged in care. Following an adjustment for variable sampling by age and gender (*Figure 12*), we found that men and women for whom sexual partners could be inferred phylogenetically contributed similarly to viral transmission of HIV-1 infection in the Botswana/Ya Tsie trial (see Materials and methods section on adjustment for variable sampling rates across different demographic groups or randomized-HIV-interventions). There would

also be a contribution to HIV incidence from individuals whose source partner resided in a community outside the trial, though not evaluated in this analysis as HIV viral sequences from communities outside the trial population were not available.

Prior to this study, a recent consensus sequence phylogenetic study assessing age-disparate sexual partnerships in Kwa-Zulu Natal, South Africa identified men older than 25 years as an important source of viral transmission to younger women aged 15–25 years. Age disparate partnerships were defined as ones with an age-gap of more than 5 years. The Kwa-Zulu Natal study was based on a community sampling fraction of 4% that was not specifically targeted to people with HIV and assumed that the direction of viral transmission within clusters would be from older partners to younger ones (*de Oliveira et al., 2017*). Phylogenetic analyses in the Botswana/Ya Tsie trial were based on an overall sampling fraction of 14% of people aged 16–64 years living with HIV in trial communities; the sampling fraction ranged from 2.7% to 36.2% among the 30 trial communities. In the phylogenetic analyses of the Botswana/Ya Tsie study, there was little genetic evidence from analyses of deep-sequence data to suggest that older men were a substantial source of HIV-1 infection to younger women in Botswana. This finding is relevant to programs aimed at reducing HIV incidence in young women in sub-Saharan Africa such as the Determined, Resilient, Empowered, AIDS-free, Mentored, and Safe women (DREAMS) partnership. Consistent with our results, the Vaginal and Oral Interventions to Control the Epidemic (VOICE) trial, a placebo-controlled randomized study of pre-exposure prophylaxis to test the efficacy of providing oral and vaginal gel tenofovir to women in Southern Africa to prevent HIV infection (*Saag, 2015*; *Balkus et al., 2015*), found no association between age-disparate relationships and risk of HIV-1 infection in young women under 25 years residing in South Africa in the cities of Durban, Johannesburg, and Klerksdorp who had male partners that were >5 years older at enrollment.

Taken together, these findings might be relevant for understanding heterogeneity of HIV transmission in similarly designed phylogenetic studies in sub-Saharan Africa based on the sample size, demographic population, and adjustments for variable sampling.

Participants with genetically similar HIV-1 infections in the Botswana/Ya Tsie trial tended to aggregate in small-sized (two- or three-person) clusters. This is consistent with phylogenetic clustering studies in generalized epidemics in Southern Africa but contrasts with larger clusters typically found in concentrated HIV epidemics in Europe and North America (*Grabowski et al., 2018*). The small-sized genetic similarity clusters identified in trial communities might reflect undersampling of key populations, for example, men and younger individuals, in the HIV transmission network. Alternatively, the predominance of smaller clusters in generalized epidemics compared with larger clusters in concentrated epidemics might reflect differences in the manner in which sexual networks under the two settings evolve over time.

Most participants in the household survey of the Botswana/Ya Tsie trial indicated that they had not engaged in transactional sex in the last 12 months. This suggested that there was a limited contribution of sex workers to the transmission patterns identified in the trial.

Our findings provide insight on HIV transmission patterns in Botswana over a 5-year period between 2013 and 2018 during which the Botswana/Ya Tsie trial was conducted. An important next step would be the integration of the large database of HIV viral genomes assembled by the Botswana/Ya Tsie trial into a real-time HIV genomic surveillance program to highlight emerging hubs of HIV spread and inform targeted studies of HIV prevention. Such an effort would be facilitated by samples routinely collected for HIV viral load monitoring and the continuously decreasing cost of high-throughput sequencing.

Our study had several limitations. First, viral genomes from an estimated 14% of all people with HIV aged 16–64 years were sampled among participating trial communities. Comparatively, the sampling fraction of phylogenetic studies based on African genomes is commonly under 10% (*Grabowski et al., 2018*). Although our sampling fraction was a modest improvement over previous studies it is a relatively small sample of HIV viral genomes in the trial population. To estimate HIV transmission flows in the trial population based on probable viral transmission events identified in the deep-sequenced sample we used the method of *Carnegie et al., 2014* in which undetected HIV transmissions were assumed to be missing at random conditional on group membership (*Carnegie et al., 2014*) (see Materials and methods section on adjustment for variable sampling rates across different demographic groups or randomized-HIV-interventions). Here group membership

refers to demographic groups (age group, gender, trial community, and geographical region) or randomized-HIV-interventions.

Of the 82 probable male-female transmission pairs identified in the deep-sequenced sample there was a single individual involved in multiple transmission events; a male who transmitted to two females each residing in different communities from his own. This suggested that only a small degree of bias in estimation of CIs would arise from a simplifying assumption of independence between inferred transmission events; and allowed us to estimate CIs using methods for multinomial proportions (see Materials and methods section on computation of CIs for estimated HIV transmission flows in the trial population).

Second, although our estimates of HIV transmission flows in the trial population were adjusted for differential sampling we cannot exclude the possibility that unmeasured factors with influence on HIV transmission flows may have impacted our results. Most participants with HIV in this trial were virally suppressed, this might have biased our results in the sense that there might be some differences between the HIV transmission patterns we found and those that exist among viral genetic clusters largely populated by people with unsuppressed virus. Such heterogeneity in HIV transmission patterns might occur due to differences in risk behavior. Nevertheless, this study was based on a relatively representative sample of adults with HIV living in rural and peri-urban villages across Eastern Botswana.

In sum, we identified several key factors in the Botswana/Ya Tsie trial that impacted HIV transmission dynamics with potential relevance to similar studies done in sub-Saharan Africa. We found that most HIV transmissions in the Botswana/Ya Tsie trial occurred between similarly aged partners within the same trial community or between trial communities in close proximity. Moreover, there was a greater flow of HIV transmissions into intervention communities from control communities than vice versa potentially reducing the observed effect-size of the trial, as would transmissions averted by the intervention that were not picked up because the recipient was not in a trial community.

We recommend widely distributed and easily accessible HIV testing (e.g., universal HIV testing campaigns), treatment, and linkage-to-care to support people as they intermix within and across communities; augmented by targeted programs that might offer pre-exposure prophylaxis to younger women and the option of self-testing for men. Population-level genomic surveillance programs to identify communities with high flows of viral transmission within and between them would be helpful in spotting emerging hubs of transmission that can be prioritized for intervention. Taken together, these measures may reduce new HIV infections and shorten the time to epidemic control.

## Code availability

Algorithms to estimate HIV transmission flows within and between population groups accounting for sampling variability and corresponding CIs have been implemented as an R package, *bumblebee* that will be made available at the following URL: https://magosil86.github.io/bumblebee. A step-by-step tutorial on how to estimate HIV transmission flows with *bumblebee* and accompanying example data sets can be accessed at: https://github.com/magosil86/bumblebee/blob/master/vignettes/bumble-bee-estimate-transmission-flows-and-ci-tutotial.md (*Magosil, 2021*).

## Acknowledgements

The authors are grateful to participants and collaborators from the Botswana Combination Prevention Project and PANGEA consortium for their support during this work. Additionally, the authors thank Susan Eshleman for contributing data from the HPTN 052 study and Roger Shapiro for his helpful comments on early versions of this manuscript. This study was supported by the National Institute of General Medical Sciences (U54GM088558), the Fogarty International Center (FIC) of the U.S. National Institutes of Health (D43 TW009610), and the President's Emergency Plan for AIDS Relief through the Centers for Disease Control and Prevention (CDC) (Cooperative agreements U01 GH000447 and U2G GH001911).

## Additional information

### Competing interests

Shahin Lockman: participates in a data safety monitoring board for NIH-funded study of PK of TB drugs and antiretrovirals in children and on a scientific advisory board for observational study of DTG programmatic rollout in Botswana. Is also a member of the Finance Board and a member of the Board of Directors for the Botswana Harvard AIDS Institute Partnership. Receives no financial compensation for these roles, and has no other competing interests to declare. Marc Lipsitch: is a Reviewing Editor for eLife. Has received consultancy fees from Merck, University of Virginia Miller Center and Janssen, and has performed unpaid consultancy work for Janssen, Pfizer and Astra Zeneca. Has also received payments or honoraria from Sanofi Pasteur and Bristol Myers Squibb. ML has no other competing interests to declare. The other authors declare that no competing interests exist.

### Funding

| Funder | Grant reference number | Author |
|--------|------------------------|--------|
| Fogarty International Center | D43 TW009610 | Lerato E Magosi |
| Centers for Disease Control and Prevention | U01 GH000447 and U2G GH001911 | Lerato E Magosi<br>Janet Moore<br>Pam Bachanas<br>Refeletswe Lebelonyane<br>Molly Pretorius Holme<br>Shahin Lockman<br>Myron Max Essex |
| National Institutes of Health | | Christophe Fraser<br>Marc Lipsitch |
| Bill and Melinda Gates Foundation | | Christophe Fraser |

The funders had no role in study design, data collection and interpretation, or the decision to submit the work for publication.

### Author contributions

Lerato E Magosi, Conceptualization, Formal analysis, Investigation, Methodology, Software, Visualization, Writing – original draft, Writing – review and editing; Yinfeng Zhang, Janet Moore, Pam Bachanas, Tebogo Segolodi, Refeletswe Lebelonyane, Sikhulile Moyo, Joseph Makhema, Resources, Writing – review and editing; Tanya Golubchik, Vladimir Novitsky, Data curation, Resources, Writing – review and editing; Victor DeGruttola, Methodology, Validation, Writing – original draft, Writing – review and editing; Eric Tchetgen Tchetgen, Methodology, Validation, Writing – review and editing; Molly Pretorius Holme, Shahin Lockman, Project administration, Resources, Writing – review and editing; Christophe Fraser, Data curation, Methodology, Resources, Validation, Writing – review and editing; Myron Max Essex, Conceptualization, Funding acquisition, Resources, Supervision, Writing – review and editing; Marc Lipsitch, Conceptualization, Funding acquisition, Methodology, Resources, Supervision, Validation, Writing – original draft, Writing – review and editing

### Author ORCIDs

Lerato E Magosi ![ORCID]http://orcid.org/0000-0002-3388-9892
Tanya Golubchik ![ORCID]http://orcid.org/0000-0003-2765-9828
Marc Lipsitch ![ORCID]http://orcid.org/0000-0003-1504-9213

### Ethics

Human subjects: The BCPP study was approved by the Botswana Health Research and Development Committee and the institutional review board of the Centers for Disease Control and Prevention; and was monitored by a data and safety monitoring board and Westat. Written informed consent for enrollment in the study and viral HIV genotyping was obtained from all participants.

### Decision letter and Author response

Decision letter https://doi.org/10.7554/eLife.72657.sa1

Author response https://doi.org/10.7554/eLife.72657.sa2

## Additional files

### Supplementary files

• Supplementary file 1. Supplemental tables 1A to O.

• Supplementary file 2. Membership and affiliations of the Botswana Combination Prevention Project and PANGEA consortium.

• Transparent reporting form

### Data availability

All relevant data are within the paper, figures and tables. HIV-1 viral whole genome consensus sequences are provided as a Dryad dataset (https://doi.org/10.5061/dryad.0zpc86706). HIV-1 reads are available on reasonable request through a concept sheet proposal to the PANGEA consortium. Contact details are provided on the consortium website (https://www.pangea-hiv.org). Code availability: Algorithms to estimate HIV transmission flows within and between population groups accounting for sampling variability and corresponding confidence intervals have been implemented as an R package, bumblebee that will be made available at the following URL: https://magosil86.github.io/bumblebee A step-by-step tutorial on how to estimate HIV transmission flows with bumblebee and accompanying example datasets can be accessed at: https://github.com/magosil86/bumblebee/blob/master/vignettes/bumblebee-estimate-transmission-flows-and-ci-tutotial.md copy archived at swh:1:rev:e44b55de833780defd37c81d2bb94f65ed1dff12.

The following dataset was generated:

| Author(s) | Year | Dataset title | Dataset URL | Database and Identifier |
|---|---|---|---|---|
| Magosi LE, Zhang Y, Golubchik T, DeGruttola V, Tchetgen Tchetgen E, Novitsky V, Moore J, Bachanas P, Segolodi T, Lebelonyane R, Pretorius Holme M, Moyo S, Makhema J, Lockman S, Fraser C, Essex M, Lipsitch M | 2022 | Deep-sequence phylogenetics to quantify patterns of HIV transmission in the context of a universal testing and treatment trial - BCPP/ Ya Tsie trial | http://dx.doi.org/10.5061/dryad.0zpc86706 | Dryad Digital Repository, 10.5061/dryad.0zpc86706 |

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
