## [Editor Report]

The study by Magosi et al., evaluates the impact of targeted public health interventions on the HIV-1 transmission rate in Botswana. Using data from a large trial in Botswana, the authors found that HIV-1 transmission was more common to occur from control population groups into targeted population groups than vice-versa. The study is of public health interest, showing how some public health interventions are powerful in reducing HIV-1 transmission but only among the population targeted. This is a very comprehensive research study showing the advantages of using deep sequencing data in combination with phylogenetic tools to assess the positive impact of public health interventions in reducing HIV-1 transmission.

---

## [Decision Letter]

**Decision letter after peer review:**

Thank you for submitting your article "Deep-sequence phylogenetics to quantify patterns of HIV transmission in the context of a universal testing and treatment trial – BCPP/ Ya Tsie trial" for consideration by *eLife*. Your article has been reviewed by 2 peer reviewers, and the evaluation has been overseen by a Reviewing Editor and Miles Davenport as the Senior Editor. The following individual involved in review of your submission has agreed to reveal their identity: Nadine Tschumi (Reviewer #1).

Essential revisions:

Overall, we think the study was well designed and using a large sample size is highly valuable. The study provides detailed information on the methods including the release of a new R package for calculating transmission flow among populations. This is highly valuable for the interpretation of public health interventions, particularly test-and-treat. However, we all agree that it is important to add some clarifications of the methods used.

1) The authors use a combination of HIV-TRACE and Cluster Picker as tools for cluster identification. As such, the authors implement a genetic threshold of 4.5% genetic diversity. This is the core outcome of the study; all the subsequent analyses are depended on the clusters identified. HIV-TRACE is often used instead of a phylogenetic tree, as the latter can be computational time-consuming with large data sets. However, the study also estimates a phylogenetic tree, thus we wish for more clarification on why both methods were used and how different the cluster outcomes are with each method.

2) The cluster analysis resulted in many infection pairs between same-sex couples, indicating missing data in the transmission chains. The study excluded these pairs from subsequent analysis, resulting in a small number of overall pairs for the final transmission flow count. We think that such 'missing link' pairs could still be included in the analysis of estimating transmission flow between geographic regions. We were wondering if the study has investigated this and believe a constructive discussion in that regard could be favorable. See comments of reviewer 2.

3) We agree that the study is overall well-presented and well-written, however, we think it would be useful to clarify some of the methods in the main text. We also agreed that the figures were overall informative and useful, however, we believe that some of the illustrations could be improved to aid with the understanding of the results. See comments of reviewer 3.

*Reviewer #1 (Recommendations for the authors):*

1. I think my concerns 8-11 above could be relatively easily addressed by sensitivity analyses showing that a change in respective definitions did not affect the main outcomes qualitatively.

2. The estimated proportion of HIV transmissions in the trial population flowing into intervention communities from control communities and vice-versa was a key finding of the manuscript. Since the number of probable transmission pairs is low, the confidence intervals are wide and overlapping. I appreciated / think that it is important and correct that you only considered probable transmission pairs for the age gap analysis. However, for this analysis I wouldn't see a problem for including pairs with probable unsampled intermediates. Was this attempted / why not?

*Reviewer #2 (Recommendations for the authors):*

The methods describing the initial sequence quality control and alignment need to be expanded. Currently, it is unclear how the shiver assembly software was run. While the sequencing and bioinformatics pipelines used here have been used in previous studies the possible impact of sequencing errors should be discussed. Ideally, replicates would be used to determine the expected rate of sequencing errors. However, other approaches such as the proportion of minority variants at different coding locations for different variant frequency cutoffs could also be used (Dyrdak et al., 2019). This is particularly important given that most trial participants were virally suppressed which could impact the overall quality of the sequencing data.

Much of the downstream analysis and results relies on the initial separation of sequenced samples into putative transmission clusters. As a result, the choice of the 4.5% genetic distance threshold and the decision to combine the results of HIV-trace and cluster-picker needs much stronger motivation. This is particularly so as Supplementary Figures 7 and 8 suggests that the two methods give quite different results. Whilst I realise cluster picker is commonly used, HIV trace does not rely on the accuracy of a provided phylogeny and as the following transmission analysis only considers pairwise relationships, HIV trace seems more appropriate.

I may have misunderstood but the method for calibrating the false discovery rate (FDR) for the inference of direct transmission described in Supplementary note 3 appears to consider all pairwise relationships (equation 6). This would only be possible if a person was able to be infected by multiple different transmitters. Assuming that the authors have only considered a single transmitter per infection then this would lead to an overestimate of the FDR. This should be clarified in the text.

Finally, it would be useful if the authors could discuss what impact high risk communities such as sex workers could have on these results and the estimates of intercommunity transmission.

– Many of the methods are described in the supplementary materials which makes the interpretation of the results challenging. The manuscript would be improved by moving some of the supplementary methods into the main text.

– The command line parameters used to run MAFFT and HIValign should be provided.

– While I was able to successfully download and install the 'bumblebee' R package I could not get a small trial example to run. Example data and a short 'how to guide' should be included as part of the package.

– Figure 7 is quite confusing and I think relatively little is gained by the use of an alluvial plot. I think it would be clearer to separate this plot into multiple bar plots of the total number of transmissions rather than the proportion within each category.

– It would be better to combine Figures 9 and 10 into a multi panel figure

– Lines 177, 184, 216, 219, 279, 342, 345 etc: The multiplication symbols look like a variable and should be removed.

– Line 262: I think this should say that the Goodman method estimates CIs for the parameters of the multinomial distribution.

---

## [Author Response]

Essential revisions:Overall, we think the study was well designed and using a large sample size is highly valuable. The study provides detailed information on the methods including the release of a new R package for calculating transmission flow among populations. This is highly valuable for the interpretation of public health interventions, particularly test-and-treat. However, we all agree that it is important to add some clarifications of the methods used.

We thank the reviewers for their interest in this novel approach that uses deep-sequence phylogenetics to improve our understanding of HIV transmission patterns in the context of the universal HIV test-and-treat intervention.

1) The authors use a combination of HIV-TRACE and Cluster Picker as tools for cluster identification. As such, the authors implement a genetic threshold of 4.5% genetic diversity. This is the core outcome of the study; all the subsequent analyses are depended on the clusters identified. HIV-TRACE is often used instead of a phylogenetic tree, as the latter can be computational time-consuming with large data sets. However, the study also estimates a phylogenetic tree, thus we wish for more clarification on why both methods were used and how different the cluster outcomes are with each method.

We thank the reviewers for their comments and are happy to provide additional results that motivate our decision to use the union of clusters detected with HIV-TRACE and Cluster Picker to estimate HIV transmissions within and between demographic sub-groups in the Botswana – Ya Tsie trial population. The primary motivation was that a filtering step was required to save time and computational resources from evaluating sequences that were too distantly related, before applying the “gold standard” of Phyloscanner to detect directed (when possible) transmission pairs. Accordingly, clustering algorithms plus a distance threshold helped to achieve this filtering. Because we shared what we take to be the reviewers’ concerns about either of the algorithms alone, we sought to maximize the number of transmission pairs that could be identified between participants in the Botswana – Ya Tsie trial with Phyloscanner by using the union of clusters detected with HIV-TRACE and Cluster Picker. This also served as a sensitivity analysis that allowed us to evaluate the extent to which the clustering patterns observed were specific to a single algorithm.

Furthermore, a previous study done by Rose and colleagues (PMID: 27824249) to compare the number and size of clusters identified with HIV-TRACE and Cluster Picker clustering algorithms revealed that HIV-TRACE generally identified larger but fewer clusters, compared with clusters identified with Cluster Picker that were typically more numerous and mostly small 2-person clusters (Please see Figure 3B in Rose and colleagues (PMID: 27824249)). This suggested that HIV-TRACE would be helpful in detecting potentially larger transmission chains and Cluster Picker would be valuable in revealing potential transmission events between pairs of individuals.

Of the 236 genetic clusters detected with the two algorithms, we identified 19 full or partial clusters (including 41 sequences) that included members that were only detected with HIV-TRACE and 122 full or partial clusters (including 242 sequences) that were unique to Cluster Picker. Moreover, of the 82 directed male-female transmission pairs inferred from the sample, (n = 5) were from genetic clusters that were unique to HIV-TRACE compared with (n = 27) that were from clusters unique to Cluster Picker. Of the five transmission events unique to HIV-TRACE clusters, three occurred in intervention communities originating from control communities. By contrast, four of the twenty-seven transmission events unique to Cluster Picker clusters occurred in intervention communities from control communities.

In summary, estimates of HIV transmissions in the trial population based on the full overlap of clusters detected with HIV-TRACE and Cluster Picker would have excluded 32 of the 82 male-female pairs used for the primary analysis. To improve clarity, we have updated the methods, results and discussion with the following statements respectively:

“Consensus sequence phylogenetics to identify clusters of participants with genetically similar HIV-1 infections. To save time and computational resources from evaluating sequences that were too distantly related, we first identified clusters of participants with genetically similar HIV-1 infections as a filtering step, before performing ancestral host state reconstruction with Phyloscanner to detect probable directed transmission pairs. Two clustering algorithms, HIV Transmission Cluster Engine (HIV-TRACE) v0.4.4 [26, 27] and Cluster Picker v1.2.3 [28], were used to identify clusters of individuals whose HIV-1 viral whole genome consensus sequences were genetically similar -- suggesting they were probably members of a transmission chain [29, 30]. HIV-TRACE defines clusters based on pairwise genetic distances only; comparatively, Cluster Picker identifies clusters using pairwise genetic distances with the guidance of a phylogenetic tree. A multiple sequence alignment (as described in the methods section on comparing genetic distances between HIV-1 viral consensus sequences of trial participants included in phylogenetic analyses) was provided as input to HIV-TRACE and Cluster Picker. Additionally, for cluster picker, a corresponding phylogenetic tree inferred with FastTree2 v2.1.10 and boot-strap support values approximated with the Shimodaira-Hasegawa test [31] were provided as inputs. We defined genetic similarity clusters as groups of two or more participants whose viral whole genome consensus sequences were separated by a genetic distance at or smaller than 4.5% nucleotide substitutions per site--and, for Cluster Picker, a bootstrap support value of at least 80%. The genetic distance threshold of 4.5% nucleotide substitutions per site was motivated by the distribution of genetic distances separating HIV-1 subtype C viral whole genomes of epidemiologically-linked couples in the HIV Prevention Trials Network (HPTN) 052 trial (Figure 6) [8, 32]. A listing of parameters used for consensus-sequence phylogenetics with HIV-TRACE and Cluster Picker is provided in Supplementary file 1 Table 1B.”

“About 1 in 7 genotyped participants included in consensus sequence phylogenetic analyses were assigned to genetic similarity clusters of people with closely related HIV-1 infections. … To maximize the number of transmission pairs that could be identified between participants in the Botswana – Ya Tsie trial with Phyloscanner we used the union of clusters detected with HIV-TRACE and Cluster Picker.”

“We first used HIV-1 whole genome viral consensus sequences to identify clusters of trial participants with genetically similar HIV-1 infections. This was done as a filtering step to save time and computational resources by excluding distantly related sequences. Thereafter, we employed deep-sequence phylogenetics to resolve the probable order of HIV transmission events within each identified cluster.”

2) The cluster analysis resulted in many infection pairs between same-sex couples, indicating missing data in the transmission chains. The study excluded these pairs from subsequent analysis, resulting in a small number of overall pairs for the final transmission flow count. We think that such 'missing link' pairs could still be included in the analysis of estimating transmission flow between geographic regions. We were wondering if the study has investigated this and believe a constructive discussion in that regard could be favorable. See comments of reviewer 2.

We thank the reviewers for their suggestion and now report in the results the findings of a sensitivity analysis as follows:

“A sensitivity analysis to evaluate the impact of probable transmission pairs with unsampled intermediates on the patterns of HIV transmission within and between intervention communities and control communities in the Botswana – Ya Tsie trial. The first set of sensitivity analyses were performed with highly supported directed same- and opposite-sex transmission pairs identified in: (1) HIV-TRACE clusters, (2) Cluster Picker clusters, (3) the overlap of HIV-TRACE and Cluster Picker clusters and (4) the union of HIV-TRACE and Cluster Picker clusters (Supplementary file 1 Table 1N). The second set of sensitivity analyses were performed for the same four categories but restricted to highly supported directed opposite-sex pairs only where the recipient in a transmission pair was first sampled about a year or more after the end of baseline household survey activities in their community (i.e. post-baseline) (Supplementary file 1 Table 1O). In both sets of sensitivity analyses the results were consistent with the primary analysis: Transmissions into intervention communities from control communities were more common than the reverse post-baseline (Supplementary file 1 Tables 1N and 1O) (see methods section on adjustment for variable sampling rates across different demographic groups or randomized-HIV-interventions). The signal was stronger as well as being more interpretable in the analysis restricted to opposite-sex pairs compared to the analysis that included same-sex pairs, which likely included one or more unsampled intermediates.”

However, it is important to note that in a predominantly heterosexual epidemic same-sex pairs might represent pairs with one or more missing intermediates or false positives. Therefore, it would be challenging to interpret the estimated HIV transmission flows in the trial population without information on the number of missing intermediates, their gender, community and trial arm. For example, consider a directed female-female pair where female-1 was sampled from an intervention community (F1-intervention) and female-2 a control community (F2-control). Let the two sampled females be connected by a single unsampled male affording the following transmission chain, F1-intervention -> M – > F2 -control. If the male belongs to a control community the true transmission flow would be from intervention-to-control and control-to-control while if the male belongs to an intervention community then it would be intervention-to-intervention and intervention-to-control. These two are very different in terms of their implications, and we have no way to distinguish them. Thus same-sex pairs were excluded from the primary analyses to limit false positives and aid interpretation of the analysis.

3) We agree that the study is overall well-presented and well-written, however, we think it would be useful to clarify some of the methods in the main text. We also agreed that the figures were overall informative and useful, however, we believe that some of the illustrations could be improved to aid with the understanding of the results. See comments of reviewer 3.

We thank the reviewers for their suggestion. To improve clarity we have moved all Supplementary Notes to the methods section and we now also provide a detailed listing of parameters used for: Shiver, MAFFT, HIValign, HIV-TRACE, Cluster Picker and FASTTREE2 in Supplementary file 1 Table 1B. Furthermore, we have replaced the alluvial plot with a barplot (Figure 16) and combined Figures that show the estimated transmission flows of HIV-1 infection in the Botswana – Ya Tsie trial population within and between geographical regions and trial arms into a single multi-panel figure (Figures 18A and 18B) in line with the reviewer’s suggestions.

Reviewer #1 (Recommendations for the authors):1. I think my concerns 8-11 above could be relatively easily addressed by sensitivity analyses showing that a change in respective definitions did not affect the main outcomes qualitatively.2. The estimated proportion of HIV transmissions in the trial population flowing into intervention communities from control communities and vice-versa was a key finding of the manuscript. Since the number of probable transmission pairs is low, the confidence intervals are wide and overlapping. I appreciated / think that it is important and correct that you only considered probable transmission pairs for the age gap analysis. However, for this analysis I wouldn't see a problem for including pairs with probable unsampled intermediates. Was this attempted / why not?

Kindly refer to the responses to essential revision numbers 1 and 2.

Reviewer #2 (Recommendations for the authors):The methods describing the initial sequence quality control and alignment need to be expanded. Currently, it is unclear how the shiver assembly software was run. While the sequencing and bioinformatics pipelines used here have been used in previous studies the possible impact of sequencing errors should be discussed. Ideally, replicates would be used to determine the expected rate of sequencing errors. However, other approaches such as the proportion of minority variants at different coding locations for different variant frequency cutoffs could also be used (Dyrdak et al., 2019). This is particularly important given that most trial participants were virally suppressed which could impact the overall quality of the sequencing data.

Kindly refer to the response to essential revision number 3. Furthermore, we agree with the reviewer that the impact of sequencing errors for sequences generated from proviral DNA is an important issue to discuss. Accordingly, we have expanded the methods as follows:

“Paired-end deep-sequencing of HIV viral genomes for phylogenetic analyses. … Moreover, the quality of sequencing was assessed with standard metrics for deep (or next-generation) sequencing data, however, we cannot exclude the potential for sequencing errors­ arising from hypermutations. The shiver sequence assembly software [20] was used to assemble and map each participant’s deep-sequencing short reads to a de-novo reference sequence tailored to the participant’s viral population. A listing of command-line parameters used to assemble HIV viral whole genomes with Shiver is provided in Supplementary file 1 Table 1B.”

Studying the impact of hypermutations on the quality of proviral DNA sequences is an interesting question that we prefer to reserve for future study, not least as it is not trivial.

Much of the downstream analysis and results relies on the initial separation of sequenced samples into putative transmission clusters. As a result, the choice of the 4.5% genetic distance threshold and the decision to combine the results of HIV-trace and cluster-picker needs much stronger motivation. This is particularly so as Supplementary Figures 7 and 8 suggests that the two methods give quite different results. Whilst I realise cluster picker is commonly used, HIV trace does not rely on the accuracy of a provided phylogeny and as the following transmission analysis only considers pairwise relationships, HIV trace seems more appropriate.

Kindly refer to the response to essential revision number 1.

I may have misunderstood but the method for calibrating the false discovery rate (FDR) for the inference of direct transmission described in Supplementary note 3 appears to consider all pairwise relationships (equation 6). This would only be possible if a person was able to be infected by multiple different transmitters. Assuming that the authors have only considered a single transmitter per infection then this would lead to an overestimate of the FDR. This should be clarified in the text.

We are happy to clarify for the reviewer that the estimated false discovery rate is indeed an upper bound as described in the results.

“Deep-sequence phylogenetics to infer the probable order of transmission events within identified clusters of genetically similar HIV-1 infections … We used the probability of inferring a phylogenetically-linked directed male-male pair in the sample to calibrate an upper bound on the false-discovery rate of inferring direct transmission between males and females in phylogenetically-linked male-female pairs; the estimated number of linked male-female pairs in the sample with unsampled intermediates was approximately 30, corresponding to a false discovery rate of 36% (30 / 82) (see methods section on estimating error rates in phylogenetic inference of direct transmission between sampled males and females). This estimated false positive rate is likely inflated given that two individuals would need to be missing from the sequenced sample to incorrectly infer transmission in a male-female pair, whereas only a single female would need to be missing to erroneously infer transmission in a male-male pair.”

In line with the reviewer’s suggestion this point is further emphasized in the methods section on estimating error rates in phylogenetic inference of direct transmission between sampled males and females as follows:

“This represents an upper bound on the false-discovery rate of inferring direct transmission between males and females in phylogenetically-linked male-female pairs.”

Finally, it would be useful if the authors could discuss what impact high risk communities such as sex workers could have on these results and the estimates of intercommunity transmission.

In line with the reviewer’s suggestions we have updated the discussion as follows:

“Most participants in the household survey of the Botswana – Ya Tsie trial indicated that they had not engaged in transactional sex in the last twelve months. This suggested that there was a limited contribution of sex workers to the transmission patterns identified in the trial.”

– Many of the methods are described in the supplementary materials which makes the interpretation of the results challenging. The manuscript would be improved by moving some of the supplementary methods into the main text.

We thank the reviewer for their suggestion and would like them to kindly note that all previous Supplementary Notes are now presented in the methods section.

– The command line parameters used to run MAFFT and HIValign should be provided.

A detailed listing of parameters used to run MAFFT and HIVAlign is now provided in Supplementary file 1 Table 1B.

– While I was able to successfully download and install the 'bumblebee' R package I could not get a small trial example to run. Example data and a short 'how to guide' should be included as part of the package.

We thank the reviewer for their comment and apologize that this was insufficiently clear in the paper. To improve clarity we have updated the code availability section and methods as follows:

“A step-by-step tutorial on how to estimate HIV transmission flows with bumblebee and accompanying example datasets can be accessed at: https://github.com/magosil86/bumblebee/blob/master/vignettes/bumblebee-estimate-transmission-flows-and-ci-tutotial.md ”.

– Figure 7 is quite confusing and I think relatively little is gained by the use of an alluvial plot. I think it would be clearer to separate this plot into multiple bar plots of the total number of transmissions rather than the proportion within each category.

Kindly refer to the response to essential revision number 3.

– It would be better to combine Figures 9 and 10 into a multi panel figure

Kindly refer to the response to essential revision number 3.

– Lines 177, 184, 216, 219, 279, 342, 345 etc: The multiplication symbols look like a variable and should be removed.

We thank the reviewer for their suggestion and have updated the manuscript accordingly.

– Line 262: I think this should say that the Goodman method estimates CIs for the parameters of the multinomial distribution.

We thank the reviewer for their suggestion and have updated the manuscript accordingly.